

# Two decades of dynamic change and progressive destabilization on the Thwaites Eastern Ice Shelf

Karen E. Alley[1], Christian T. Wild[2], Adrian Luckman[3], Ted A. Scambos[4], Martin Truffer[5], Erin C. Pettit[2], Atsuhiro Muto[6], Bruce Wallin[7], Marin Klinger[7], Tyler Sutterley[8], Sarah F. Child[5], Cyrus Hulen[9], Jan T. M. Lenaerts[10], Michelle Maclennan[10], Eric Keenan[10], Devon Dunmire[10]

[1]Centre for Earth Observation Science, Department of Environment and Geography, University of Manitoba, Winnipeg, MB, Canada
[2]College of earth, Ocean, and Atmospheric Sciences, Oregon State University, Corvallis, OR, USA
[3]Department of Geography, Swansea University, Swansea, UK
[4]Earth Science and Observation Center, Cooperative Institute for Research in Environmental Sciences, University of Colorado Boulder, Boulder, CO, USA
[5]Geophysical Institute and Department of Physics, University of Alaska Fairbanks, Fairbanks, Alaska, USA
[6]Department of Earth and Environmental Science, Temple University, Philadelphia, PA, USA
[7]National Snow and Ice Data Center, University of Colorado, Boulder, CO, USA
[8]Polar Science Center, University of Washington, Seattle, Washington, USA
[9]Department of Earth Sciences, College of Wooster, Wooster, OH, USA
[10]Department of Atmospheric and Oceanic Sciences, University of Colorado Boulder, Boulder, CO, USA

*Correspondence to*: Karen E. Alley (karen.alley@umanitoba.ca)

**Abstract.** The Thwaites Eastern Ice Shelf (TEIS) buttresses the eastern grounded portion of Thwaites Glacier through contact with a pinning point at its seaward limit. Loss of this ice shelf will promote further acceleration of Thwaites Glacier. Understanding the dynamic controls and structural integrity of the TEIS is therefore important to estimating Thwaites' future sea-level contribution. We present a ~20-year record of change on the TEIS that reveals the dynamic controls governing the ice shelf's past behavior and ongoing evolution. We derived ice velocities from MODIS and Sentinel-1 image data using feature tracking and speckle tracking, respectively, and combined these records with ITS_LIVE and GOLIVE velocity products from Landsat 7 and 8. In addition, we estimated surface lowering and basal melt rates using the REMA DEM in comparison to ICESat and ICESat-2 altimetry. Early in the record, TEIS flow dynamics were strongly controlled by the neighboring Thwaites Western Ice Tongue (TWIT). Flow patterns on the TEIS changed following the disintegration of the TWIT in ~2008, with a new divergence in ice flow developing around the pinning point at its seaward limit. Simultaneously, the TEIS developed new rifting that extends from the shear zone upstream of the ice rise and increased strain concentration within this shear zone. As these horizontal changes occurred, sustained thinning driven by basal melt reduced ice thickness, particularly near the grounding line and in the shear zone area upstream of the pinning point. This evidence of weakening at a rapid pace suggests that the TEIS is likely to fully destabilize in the next few decades, leading to further acceleration of Thwaites Glacier.



## 1 Introduction

Thwaites Glacier in West Antarctica holds the most important control on global sea-level rise over the next few centuries (Scambos et al., 2017). The broad causes and implications of the destabilization of Thwaites have been

understood for decades: increased delivery of warm modified Circumpolar Deep Water (mCDW) to grounding zones triggers retreat of an ice sheet grounded well below sea level (e.g. Holland et al., 2019), leading to dynamic instability and greatly accelerated ice discharge into the ocean (Hughes, 1981; Mercer, 1978, Weertman, 1974). Recent evidence suggests that the predicted irreversible retreat of Thwaites Glacier is already underway (Joughin et al., 2014; Rignot et al., 2014). However, knowing the details of the timing, magnitude, and pace of the collapse

of Thwaites are essential for more detailed forecasting of its sea-level contribution.

To understand these changes, we need to define both the oceanic forcing responsible for initiating retreat and the dynamic response that governs the inherent instability of the system. At the interface of this forcing and dynamic response are the floating ice components that form the seaward terminus of Thwaites Glacier. Because this ice

interacts directly with ocean water, changes in its velocity and thickness may reveal clues about ocean forcing (e.g. MacGregor et al., 2012; Miles et al., 2020; Pritchard et al., 2012). Ice shelves and ice tongues also actively impact the dynamic stability of the system, as contact with the seafloor and embayment walls transmits backstress to grounded ice and slows ice flow and retreat (e.g. Dupont and Alley, 2005; MacGregor et al., 2012; Reese et al., 2017). Changes in ice-shelf dynamics and surface features may therefore signal fundamental imbalances in the

system that can trigger rapid future change.

Thwaites Glacier has two floating ice areas: the Thwaites Western Ice Tongue (TWIT), and the Thwaites Eastern Ice Shelf (TEIS; Figure 1). Most of the ice discharge from Thwaites passes through a fast-flowing channel that feeds the TWIT, which is an unconfined floating ice tongue that has largely disintegrated in recent years. Until the

early 2000s, the TWIT was grounded on a subsea ridge near the ice edge (Rignot, 2001), which was likely the site of the main grounding line for this section of the ice shelf decades to centuries ago (Tinto and Bell, 2011). By 2009, the TWIT had largely lost contact with this pinning point (MacGregor et al., 2012; Tinto and Bell, 2011), although some grounding of the TWIT on the subsea ridge may have occurred intermittently for several more years (Miles et al., 2020).




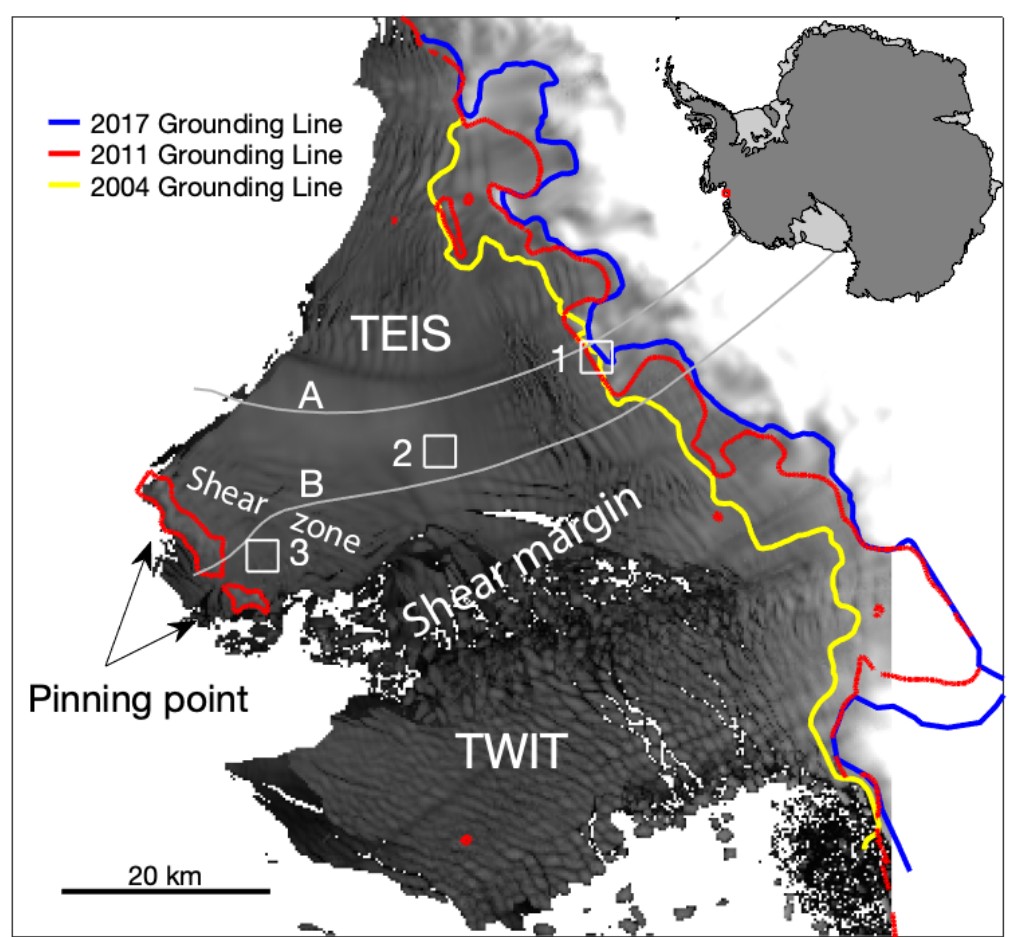

Figure 1: Location map. The Thwaites Eastern Ice Shelf (TEIS) and Thwaites Western Ice Tongue (TWIT) are labelled. We also indicate the "pinning point," the "shear zone" upstream of the pinning point, and the "shear margin" between the TEIS and the TWIT, which are terms discussed in the text. Three 3 km x 3 km sites of interest are shown, which are referred to in the text as the "grounding zone" (site 1), "mid-shelf" (site 2) and "pinning point" (site 3) areas. Data from these sites are shown in Figures 2 and 5. Flowlines based on 2015-2020 velocities, labelled A and B, are represented in Hovmöller diagrams in Figures 3 and 6. Grounding lines are from approximately 2004 (Bindschadler et al., 2011), 2011 (Rignot et al. 2011), and 2017 (Milillo et al., 2019). Figure created using the Antarctic Mapping Tools for Matlab (Greene et al. 2017).





Variability in TWIT velocity and structural integrity has been documented in detail (Miles et al., 2020; Mouginot et al., 2014). The last 20 years included periods of relatively stable velocity, which were accompanied by a
strengthening of the shear margin between the TWIT and the TEIS. However, the more recent record has been dominated by periods of instability, with increasing velocities, extremely rapid ice-edge retreat, and a loss of coherence in the TWIT/TEIS shear margin (Miles et al., 2020; Mouginot et al., 2014). As the ice tongue nearly completely detached starting around 2008, the TWIT is unlikely to return to a stable configuration with a strong TWIT/TEIS shear margin.


The large changes observed on the TWIT have also significantly impacted the behaviour of the TEIS (Miles et al., 2020; Mouginot et al., 2014), which maintains a very different configuration than the TWIT. The same ridge that pinned the TWIT to the seafloor in past decades is responsible for a large ice rumple that confines the seaward limit of the TEIS (Tinto and Bell, 2011). As this ice rumple provides significant buttressing to the grounded ice
upstream (Fürst et al., 2016; Reese et al., 2017), we will refer to it as a pinning point. This pinning point is at least partially responsible for the slower velocities and more stable calving-front positions of the TEIS as compared to the TWIT. The loss of this buttressing due to disintegration of the TEIS would therefore likely cause a step increase in ice discharge through the eastern portion of the ice stream, leading to ocean circulation changes and a response in the pace of grounding-line retreat.

In this study, we present detailed records from the last two decades of dynamic change on the TEIS. Patterns of ice-shelf speed, flow direction, and surface strain rates derived from optical and radar imagery are analysed to understand the dynamic trends and the forcings that control those trends. Data from satellite-derived DEMs and laser altimetry reveal spatial patterns of thinning across the ice shelf, which suggest details of decadal-scale ocean forcing. With the additional context of surface-feature change, our data suggest that the TEIS has exhibited
evidence of destabilization over the last two decades that is likely to continue to progress in the future.

## 2 Data and methods

### 2.1 Velocity and strain-rate data and methods

We assembled two velocity records for this analysis: a long-term (20-year) record of two-year composites of velocity maps, temporally centred on summers and derived from MODIS, Landsat-7, and Landsat-8 optical image
pairs; and a short-term (five-year) record of seasonal average velocity derived from MODIS, Landsat-8, and Sentinel-1 radar imagery. All velocities were generated by feature or speckle tracking. We also used the calculated velocities to derive flow direction and strain-rate component maps.



MODIS-based velocity estimates used in this study were derived using the Python-based image cross-correlation

software PyCorr (Fahnestock et al., 2016). MODIS images are available at 250 m spatial resolution through the
NSIDC Ice Shelf Image Archive (Scambos et al., 1996) from 2000-2019, and these images are the main source of
the velocity record presented here from 2000-2013. MODIS correlations were limited to image pairs with a
separation of at least 50 days, as the low spatial resolution requires large feature displacements for accurate
measurement. This low spatial resolution also means that MODIS correlations are inaccurate above the grounding

line, where surface features that move at the ice flow velocity are too small for MODIS to track accurately. In
these grounded areas, the features with strong correlation are primarily the surface undulations arising from ice
interaction with bedrock, yielding incorrect near-zero speeds. On floating ice, MODIS successfully correlates
larger crevasse features, basal crevasses, and rifts, and results match very closely with velocities estimated from
other sources. MODIS-derived velocity data provide most of the measurements available before 2013.


When available during the 2000-2013 time period, we have also utilized velocities derived from Landsat-7
available through the ITS_LIVE global ice velocity project (Gardner et al., 2019), but these data are severely
limited by the scan-line correction malfunction that caused significant data loss in Landsat-7 images after 2003.
Landsat-8 imagery is available from 2013, and every available image pair is processed with PyCorr and distributed

at 300 m resolution as part of the GOLIVE project (Scambos et al., 2016). We used all available correlations for
10 Landsat-8 paths/rows that overlap the TEIS. Because Landsat-8 has a higher spatial and radiometric resolution
than MODIS (15 m pixels and 12-bit digitization), correlations are successful with shorter time separations. In
areas with fewer large surface features, the algorithm applied to Landsat-8 image pairs can detect the displacement
of persistent sastrugi fields on the ice-shelf surface.


For both MODIS and Landsat-8, PyCorr was used to produce velocity correlations as well as images that describe
the correlation strength for each pixel and the difference in correlation strength between successful correlations
and neighbouring options. Velocity output images were filtered using thresholds on these parameters, which were
individually tuned according to the noise in each composite velocity grid. The results were smoothed using a 3x3

median filter to remove spurious correlations.

Despite having multiple data sources, data gaps are common early in our 20-year record. We therefore produced
each annual image by combining two full years of data centred on a summer season. Velocity correlations for each
time period were spatially interpolated to a common grid at 500 m resolution. The images were then stacked, and

a derived image for each time period was produced by taking the median value of the stack of values at each grid
cell. Small data gaps (<~5 pixels in any dimension) were filled using bilinear interpolation. The x- and y-



component velocity images were then used to calculate flow directions, as well as flow-oriented longitudinal, transverse, and shear strain rates. These strain rates were calculated using a logarithmic formulation and a 5 km length scale, which is approximately consistent with viscous processes (Alley et al., 2018).


We also produced velocity grids with seasonal temporal resolution for the last ~5 years of the record, with winter velocity values provided by radar imagery. Sentinel-1 radar imagery is available starting in late 2014, with more consistent coverage available from September 2016 with the Launch of Sentinel-1B. Velocities from Sentinel-1 were derived using feature tracking between 12-day Interferometric Wide (IW) image pairs from 2014 to

September 2016 and 6-day and 12-day image pairs between September 2016 and December 2020. We used feature tracking patch sizes of 416x128 pixels (~1-km square on the ground) and sampled every 50x10 pixels (~100 m on the ground). Feature tracking uses the Gamma Software and utilises physical features on the ice (crevasses, icebergs etc.) as well as speckle patterns where the images are phase-coherent (speckle tracking). Sentinel-1 velocity grids were filtered using the signal-to-noise ratio and an area-based noise filter and combined to produce

mean quarterly velocity maps. We utilize a record in this study derived only from Sentinel-1 imagery starting in 2014 that provides high-spatial-resolution information despite data gaps. We also produced a smoother but lower-resolution combined 5-year record with MODIS and Landsat-8 correlations. Like our 20-year record, this record was gridded at 500 m and used to calculate strain rates on a 5 km length scale.

Uncertainty in velocity estimates comes from two main sources: errors in geolocation of the satellite imagery and errors in cross-correlation. Cross-correlation errors in PyCorr are expected to be less than 0.1 pixels (Fahnestock et al., 2016), which is 25 m for MODIS imagery and 1.5 m for Landsat-8. MODIS geolocation accuracy is better than 50 m (Wolfe et al., 2002), and Landsat-8 geolocation accuracy is better than 15 m (Fahnestock et al., 2016). MODIS imagery was correlated with no less than 50-day separations between images, with most separations

between ~60 and 200 days. This yields a total maximum error estimate of ~450 ma$^{-1}$ on an ice shelf flowing at ~750 ma$^{-1}$. Landsat-8 error estimation with a minimum of 16-day separations (most were 16 to 128 days) yields an error of ~400 ma$^{-1}$. By a similar analysis, errors in Sentinel-1 velocities are estimated to be less than 100 ma$^{-1}$. These are maximum error values. More typical geolocation errors are half of the stated maximums, and with ~100 day separations, errors for a single velocity pair are ~90 ma$^{-1}$ and 25 ma$^{-1}$ for MODIS and Landsat, respectively.


In addition, these error estimates refer to individual image pairs, and our composite products stack as many image pairs as were available during each time period, taking the median value for each pixel. Assuming a normal distribution of error, this significantly increases the accuracy and precision of our velocity estimates. To get an empirical estimate of our measurement uncertainties, we calculated the uncertainty as:





$$\delta = \frac{st}{\sqrt{n-1}}$$

Where $\delta$ is the uncertainty, $s$ is the standard deviation of the pixel stack, $t$ is calculated from the standard t-distribution, and $n$ is the number of pixels in the stack. We used standard error propagation principles to estimate the uncertainty in derived flow directions and strain rates, which are shown as error bars in Figures 2 and 5.

**2.2 Surface elevation data**

Surface-elevation change was calculated using a combination of photogrammetry-derived digital elevation models (DEMs) and laser altimetry data. The Reference Elevation Model of Antarctica (REMA; Howat et al. 2019) was created using sub-meter-scale DEM strips derived from GeoEye and Worldview satellite imagery. We used a tile from the 8-m mosaicked product, which includes data from the 2013-2014 summer season in the TEIS area. The

DEM strips used to create this product were vertically referenced using Cryosat-2 altimetry data, which were projected using area-averaged thinning rates to the time that each strip was collected. Estimated elevation errors provided with the REMA tile, which take into account DEM strip creation errors and vertical referencing, are on average ± 6 m in this tile. However, the altimetry data used for referencing were not corrected for tides. Tidal amplitude in this region is approximately ± 1 m (Padman et al., 2002). As errors in DEM strip creation and vertical

referencing are uncorrelated with tides, our total estimated vertical error associated with the REMA tile is approximately ± 6 m. While this is a significant absolute error, REMA strips have been registered vertically where Cryosat-2 data are available, and nearby strips have then been referenced to each other. We therefore expect the error in REMA to be strongly spatially correlated, particularly within mosaicked tiles, allowing us to analyse spatial patterns with more confidence than absolute changes. We expect to find the largest errors at strip boundaries

where blending techniques have been used to match DEM strip edges (Howat et al. 2019).

The ICESat and ICESat-2 data were corrected following Smith et al. (2020) and Paolo et al. (2016). Data corrections were performed using the Python-based Cryosphere Altimetry Processing Toolkit (Captoolkit; https://github.com/fspaolo/captoolkit). All ICESat data were downloaded from the GLA12 release 634 data

product (Zwally et al. 2014). We applied corrections for the Gaussian-centroid offset, as well as corrections for inter-mission laser bias and signal saturation (Borsa et al., 2014). In addition, we applied filters based on several data quality flags (we retained points with use_flg = 0, sat_corr_flg < 3, att_flg =\= 0, and num_pk =1), and retained only points unaffected by clouds (cloud_flg = 0). We converted all measurements to the WGS84 ellipsoid. ICESat-2 data were provided as part of the ATL06 land-ice data release (Smith et al., 2019), which gives surface elevations

with respect to the WGS84 ellipsoid. Data were removed if they were flagged by the provided quality summary flag (atl06_quality_summary), and points were removed that were in segments with high along-track variability



or that listed unrealistic surface heights (which are most likely the result of atmospheric scattering). For both datasets, we removed the ocean tide and ocean loading corrections applied to the data in the release. We then re-tided the data with ocean tides derived from the Circum-Antarctic Tidal Simulation (CATS2008; Padman et al.,

2008), load tides from the fully global barotropic assimilation model (TPXO9) from Oregon State University developed by (Egbert and Erofeeva, 2002), and accounted for the inverse barometric effect (IBE; Dorandeu and Traon, 1999; Mathers, 2002) using sea-level pressure data from the ERA-5 reanalysis (Bell et al., 2020). ICESat and ICESat-2 points are expected to have an accuracy better than 5 cm with a precision better than 15 cm (Brunt et al., 2019).


As ocean tides, ocean loading, and IBE are generated by ocean processes, we did not apply these corrections to grounded pixels. The TEIS has experienced extensive grounding-line retreat during the past two decades. While annual estimates of grounding-line location are unavailable, we were able to obtain three grounding-line products that were used to determine floating areas in this analysis. For the ICESat data, we used the continent-wide

grounding line estimated by Bindschadler et al. (2011). This grounding line was derived using Landsat-7 data from 1999-2003 and ICESat data from 2003-2008. For our ICESat-2 data, we used the InSAR-derived 2017 grounding line location from Millillo et al. (2019), which was the most recent estimate available to us. Neither dataset includes the grounding line for the pinning point at the seaward limit of the TEIS. We therefore used a 2011 grounding line from the MEaSUREs dataset (Rignot et al., 2016) to estimate the grounded area for both DEMs. We combined

this grounding line information with BedMachine ice thickness (Morlighem et al., 2020) to create an "alpha" map for each time period (Han and Lee, 2014; Wild et al., 2019), which shows whether each pixel is freely-floating (a value of 100%) or fully grounded (a value of 0%). These maps of tide-deflection ratio were calculated with a two-dimensional elastic finite-element model, as formulated by Walker et al. (2013). Corrections for ocean and load tides and IBE were then scaled according to the percentage indicated in the alpha map before being applied to the

ICESat and ICESat-2 data. We assumed that solid Earth displacement due to ocean tidal loading was negligible above the grounding line. Comparisons of data from in situ GPS units deployed since the 2019-2020 season and the CATS2008 tide model, with load tides and IBE included, show an error of ± 17 cm in the TEIS region.

Overall, we estimated the error in the surface elevation change data to be the sum of the errors in the individual

measurements divided by the time difference between the measurements, which yields a total average error of approximately ± 1.25 m/yr for the surface lowering estimate between REMA and ICESat-2, and ± 0.75 m/year for the estimate between ICESat and REMA. We note broad agreement in the thinning patterns between the ICESat/REMA and REMA/ICESat-2 estimates, which suggests that the actual error is typically below the change





signal, and smaller than the estimates given here. We expect the largest errors to be found in areas where mosaicked
REMA strips join, with more reliable estimates within the boundaries of individual REMA strips.

**2.3 Lagrangian estimates of thickness change and basal melt**

Measurements of surface lowering and ice-thickness change, along with derived estimates of ice-shelf thinning
and basal melt rates, are most easily calculated from altimetry data using an Eulerian framework, which considers
measurements in a fixed reference frame relative to the geoid. This approach often yields large positive and
negative values that are the result of advection of ice of differing thickness, rather than representing true change
in the thickness of the ice shelf over time. We therefore used a Lagrangian framework, which calculates change in
a reference frame moving with ice flow.

To calculate Lagrangian ice-parcel flow paths, we used our annual velocity composites to migrate the altimetry
points from ICESat and ICESat-2 to the locations the ice parcels would have been when the REMA data were
collected. Velocity vector components were interpolated in both space (using bilinear interpolation) and time
(using linear interpolation) to match the time and location the altimetry points were collected. The points were
then allowed to move according to the interpolated velocity components for a time step of 10 days, at which point
interpolation was repeated. This process was continued until the points reached the same time that the REMA
pixels were collected. ICESat and ICESat-2 elevation values were smoothed along track using a moving average
over 500 m to match the resolution of the velocity measurements.

We assessed both change in surface elevation and change in ice thickness. Lagrangian surface-elevation change
(Dh/Dt) is valid on both grounded and floating ice, and was found by subtracting the surface height at the earlier
time from the surface height at the later time at migrated altimetry point locations. Ice thickness and basal melt
rates were estimated using an assumption of hydrostatic equilibrium. For these calculations, we used the alpha
maps described above to remove any ICESat or ICESat-2 points outside of hydrostatic equilibrium before
Lagrangian trajectory calculations. Following parcel movement, we removed any points that ended outside of
hydrostatic equilibrium using an alpha map based on the MEaSUREs 2011 grounding line (Rignot et al., 2011).
ICESat, ICESat-2, and REMA elevations were converted to ice thickness using (Jenkins and Doake, 1991):

$$Z_s = \left(1 - \frac{\rho_i}{\rho_w}\right) H + \left(\frac{\rho_i}{\rho_w}\right) h_a$$

where $Z_s$ is the surface elevation, $\rho_i$ is the density of ice (917 kg/m3), $\rho_w$ is the density of seawater (1026 kg/m3),
$H$ is the ice thickness, and $h_a$ is equivalent firn-air column thickness. We derived $h_a$ using a one-dimensional firn





model (SNOWPACK; Keenan et al., 2021 accepted) that is adapted for Antarctic climate conditions and forced
by MERRA-2 reanalysis (Gelaro et al., 2017). Using SNOWPACK, we simulated the evolution of a 100-m firn
column at 75°S, 106.25°W from January 1, 1980 to December 31, 2019. The model outputs % air in each firn

layer which is multiplied by layer thickness (m) and summed across all layers to obtain $h_a$.

To calculate basal melt rates, we used solid-ice-equivalent column heights, which were found by subtracting the
firn-air column thickness from the total thickness. The Lagrangian thickness change of a parcel (DH/Dt) was
calculated by differencing the ice thicknesses at migrated altimetry point locations. We then calculated basal melt
rate ($\dot{m}_b$) using mass conservation (Jenkins and Doake, 1991):


$$\frac{DH}{Dt} + H(\dot{\epsilon}_{lon} + \dot{\epsilon}_{trans}) = \dot{m}_s + \dot{m}_b$$

Where $\dot{\epsilon}_{lon}$ and $\dot{\epsilon}_{trans}$ are time-averaged longitudinal and transverse strain rates. The surface mass balance $\dot{m}_s$
was estimated using MERRA-2 atmospheric reanalysis precipitation minus evaporation and sublimation (Gelaro
et al., 2017). Using standard uncertainty propagation equations, we estimate an uncertainty of 1.35 m/yr for basal
melt rates calculated between ICESat and REMA, and 0.9 m/yr for basal melt rates calculated between REMA

and ICESat-2. Most of the uncertainty comes from the error in the REMA DEM and, as noted above, we expect
the highest uncertainty magnitudes at locations within the tile where REMA strips were feathered, and uncertainty
magnitudes are lower than the values estimated here across most of the REMA tile.

**3 Results**

**3.1 Twenty-year velocity and strain-rate records**

We analysed the twenty-year velocity record at three scales: by calculating changes in small, fixed areas of interest,
using Hovmöller diagrams to assess change along flowlines, and through annual composite maps that show
patterns over the entire shelf. For our small areas of interest, we chose three square sites covering 9 km² in regions
of the TEIS that behave in different ways (Fig. 1): site 1 crosses the 2011 grounding zone (Rignot et al., 2016),
site 2 represents mid-shelf patterns, and site 3 is just upstream of the pinning point that constrains the ice shelf.


Figure 2 shows average values of ice-flow speed, direction, and longitudinal strain rate at the three sites. The
change in ice speed over time yields the most consistent patterns in these different areas of the shelf, with all three
showing a prominent peak in speed between 2005 and 2007. Following this peak, the grounding zone and mid-
shelf sites display a small but steady increase in speed to the end of the record, while the pinning point site

experiences more variability, with an increase in speed only in the last four years of the record.





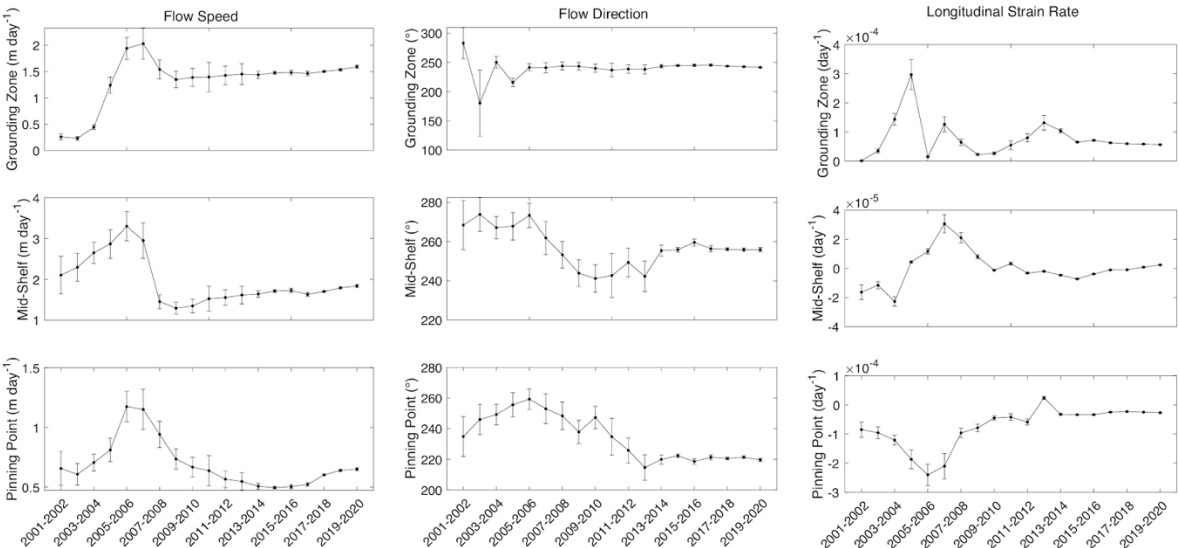

**Figure 2:** Average values of speed, flow direction, and longitudinal strain rate at the three sites of interest (locations shown in Figure 1) for the 20-year velocity record.

Flow directions are presented in grid directions based on the WGS84 Antarctic Polar Stereographic projection (EPSG:3031) used in all figures in this study. Grid north is 0˚, with values increasing clockwise. Following noise early in the record, flow directions at the grounding zone site are extremely stable. The mid-shelf and pinning point sites show more variability, but both sites show an overall decrease in angle over time, with most of the decrease concentrated in the middle of the record, coincident with the large speed decrease seen in all three boxes. This means that flow directions at these two sites shifted from grid west (270˚) or just south of grid west to a direction closer to grid south (counter-clockwise) over time.

Longitudinal strain rates show greater contrast between the shelf areas. The peak in 2004-2005 at the grounding zone site primarily reflects the retreat of the grounding line across this area. Because the MODIS-derived velocities (which are the primary data source early in the record) incorrectly show near-zero ice-flow speeds above the grounding line, the data yield artificially high longitudinal strain rates near the current grounding-line location. While this does not reflect accurate strain rates, it does give an estimate of grounding-line location throughout the record.



Patterns of longitudinal strain rate show approximately opposite trends early in the record for the mid-shelf and pinning point sites. Between 2005 and 2007, coincident with the large speed increase noted in all boxes, the mid-

shelf site shows anomalously positive (extensional) longitudinal strain rates, while the pinning point site shows anomalously negative (compressional) strain rates. Following these anomalies, longitudinal strain rates in these boxes are approximately stable, but with a slight increasing trend. Longitudinal strain rates at the mid-shelf site switch from negative (compressional) to positive (extensional) in the last two years of the record, although the difference is very small.


To provide some spatial context for the observed patterns in these areas of interest while easily visualizing change throughout the full record, we utilized Hovmöller diagrams along two flowlines of interest (grey solid lines in Figure 1). These flowlines were generated based on Sentinel-1 data averaged between 2014 and 2020. Flowline A starts above the grounding line and flows through the main calving face of the TEIS towards grid north, while

Flowline B starts above the grounding line and crosses the pinning point that confines the TEIS. The MEaSUREs 2011 grounding line (Rignot et al., 2016) is marked using vertical white lines on the Hovmöller diagrams in Figure 3. Note that the abrupt transition in velocity and strain rate near the grounding line during the first ~12 years of the record is an artefact of MODIS's inability to measure ice flow on grounded ice.

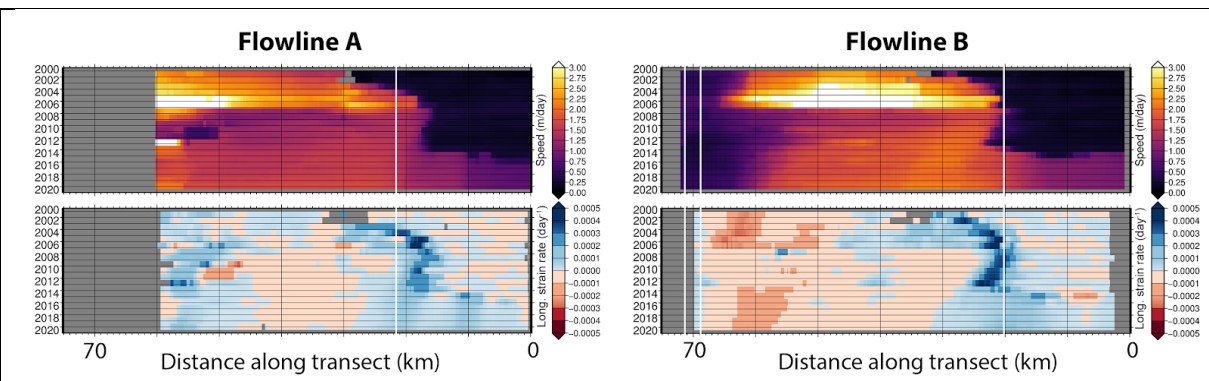

**Figure 3:** Hovmöller diagrams of speed and longitudinal strain rates from our long-term record along flowlines A and B (Figure 1). Vertical white lines represent the location of the Measures 2011 grounding line (Rignot et al., 2016).






The speed records in Figure 3 also show the increase in ice speed from the beginning of the record until ~2007 as
noted in the sites of interest examined in Figure 2. This acceleration stretches from the grounding zone all the way
to the calving front along Flowline A. The area of increased speed was confined to the region between the
grounding zone and the pinning point on Flowline B, but it migrated towards the pinning point over time before
the floating ice shelf decelerated drastically in 2007. Both flowlines show fairly small but uniform increases in
velocity following the slowdown in 2007, a trend that is consistent along the full length of the flowlines. Similar
to the more drastic increase in velocity between 2000 and 2007, this acceleration during the second half of the
record migrates towards the pinning point along Flowline B.

Longitudinal strain rates are represented as positive in extension (blue) and negative in compression (red).
Extensional strain rates near the grounding zone are artificially high prior to 2013 in our mapping because MODIS
data comprise most of the record. However, valid high longitudinal strain rates are indicated with the accurate
grounded ice velocities determined from Landsat-8 pairs from 2013 onward, representing the flow acceleration as
ice transitions from grounded to floating. Strain rates just downstream of the grounding zone were most extensive
during the 2000-2007 acceleration. Extensional strain rates are also found near the calving front along Flowline A
throughout the record. Otherwise, longitudinal strain rates are primarily compressional on the TEIS, particularly
in the shear zone in front of the seaward pinning point. During the latter part of the record, the zone of
compressional strain rates in the pinning point shear zone narrow and migrate towards the pinning point.

Patterns of change are captured in yet more spatial detail by examining maps of each variable. Videos that show
maps of speed, flow direction, and strain-rate components (longitudinal, transverse, and shear) alongside MODIS
imagery representative of each season are available at the US Antarctic Program Data Center (Alley et al. 2021).
We highlight key frames from these videos in Figure 4, including panels from early in the record (2001-2002),
during the large acceleration event (2005-2006), and late in the record (2018-2019). These spatial patterns, along
with the change in our site examples and the Hovmöller diagrams, are discussed in Section 4.



**Figure 4:** Annual maps of TEIS variables. Black dashed ovals mark features discussed in the text.





## 3.2 5-year velocity and strain-rate record

In addition to the 20-year velocity record, we also produced a shorter-term, higher-temporal-resolution velocity record from 2015-2020. For each variable, we produced four averages per year: spring (September, October, November), summer (December, January, February), fall (March, April, May) and winter (June, July, August). The winter averages are primarily derived from Sentinel-1 radar data, as visible-band images are not available

during polar winter, while the summer images combine both Sentinel-1 and visible-band images from Landsat-8 and MODIS.

Figure 5 shows speed and longitudinal strain rates from the 5-year record averaged within the same study sites identified in Figure 1. The trends in Figure 5 are consistent with the long-term record trends shown in Figure 2,

with increases in speed in all three boxes and more variability in the longitudinal strain rates. Notably, TEIS ice dynamics at these sites show no seasonal cycle.

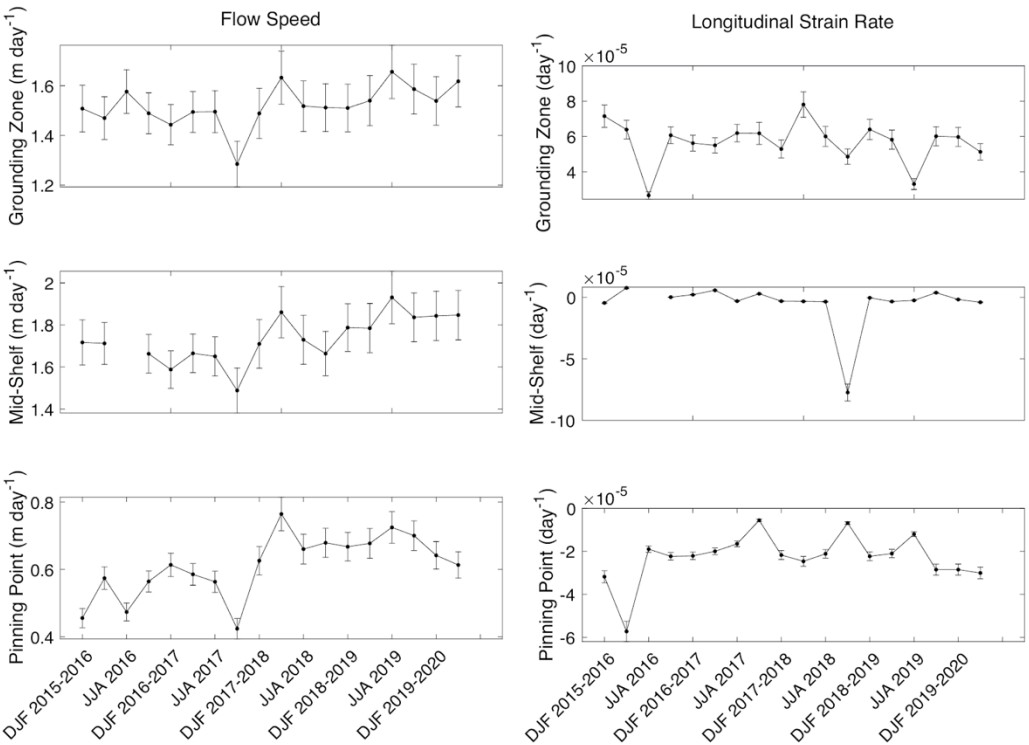

**Figure 5:** Average values of speed, flow direction, and longitudinal strain rate at the three sites of interest (locations shown in Figure 1) for the 5-year velocity record.





Considerably more detail can be seen in Hovmöller diagrams in Figure 6, which display data from the same

flowlines used in Figure 3. Figure 6 provides data from our combined 5-year record, as well as from a monthly

record based only on Sentinel-1 data. This Sentinel-1 record is at both a higher spatial (100 m) and temporal

(monthly) resolution than our combined 5-year record. In addition, strain rates are calculated on a shorter (200-m)

length scale, rather than on the longer, approximately viscous (5 km) length scale used in our combined record.

The Sentinel-1 data are therefore more appropriate for looking at details of change over small spatial length scales,

such as in the shear zone upstream of the TEIS pinning point, as they preserve sharp gradients in dynamic

properties.

The migration of higher speeds towards the pinning point seen in the 20-year record is particularly evident in the

Sentinel-1 record. Furthermore, the strongly negative longitudinal strain rates in this shear zone, which appear

constant across it in the combined 5-year record, are seen to be concentrated in three distinct bands in the Sentinel-

1 record, which we have marked with three black arrows. Two of these bands converge at the end of the record.

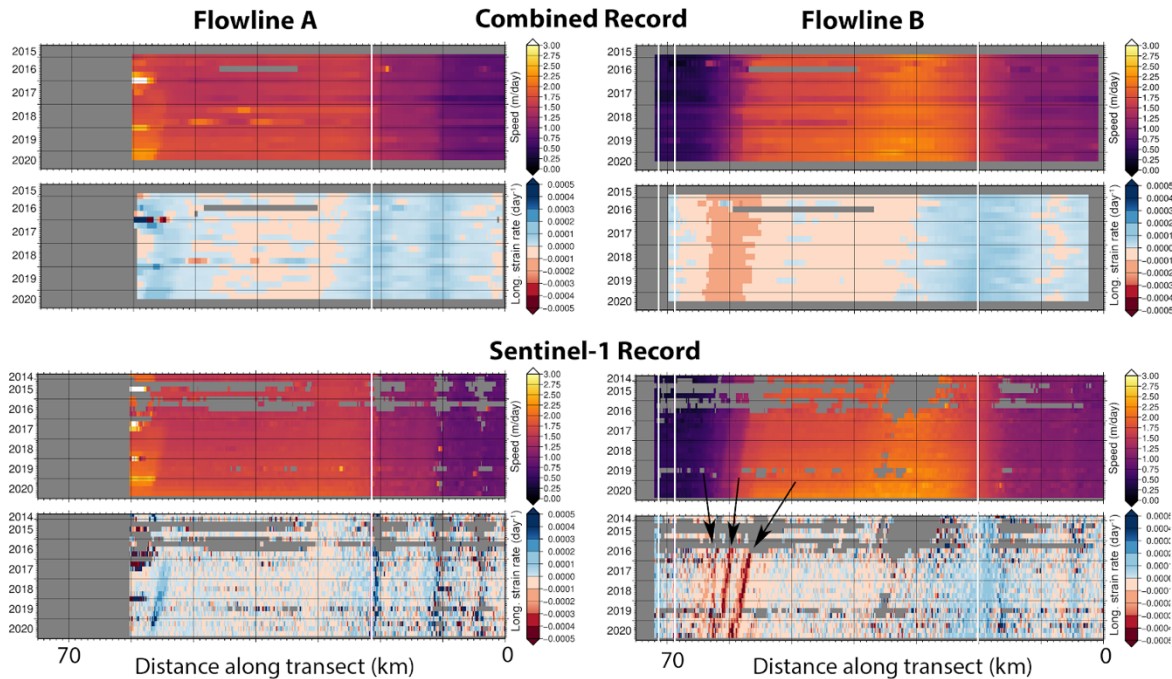

**Figure 6:** Hovmöller diagrams of speed and longitudinal strain rates from our combined 5-year record and

from Sentinel-1 radar speckle tracking for flowlines A and B (Figure 1).



### 3.3. Surface elevation change and basal melt rates

Figure 7 shows surface height change and basal melt rates calculated based on mass conservation on the TEIS.

The left-hand column (panels a and c) shows change between ICESat (data points collected between 2003-2009) and REMA (DEM strips collected between 2013-2014), and the right-hand column (panels b and d) shows change between REMA and ICESat-2 (data points collected between 2018-2020). The first row (panels a and b) gives Lagrangian surface height change, while the second row (panels c and d) gives calculated basal melt rates for pixels on the freely floating ice shelf. All points are plotted with ICESat or ICESat-2 points migrated to their

locations when the REMA data were collected. Surface height change and basal melt are calculated as annual averages over the time periods represented by each set of data points.

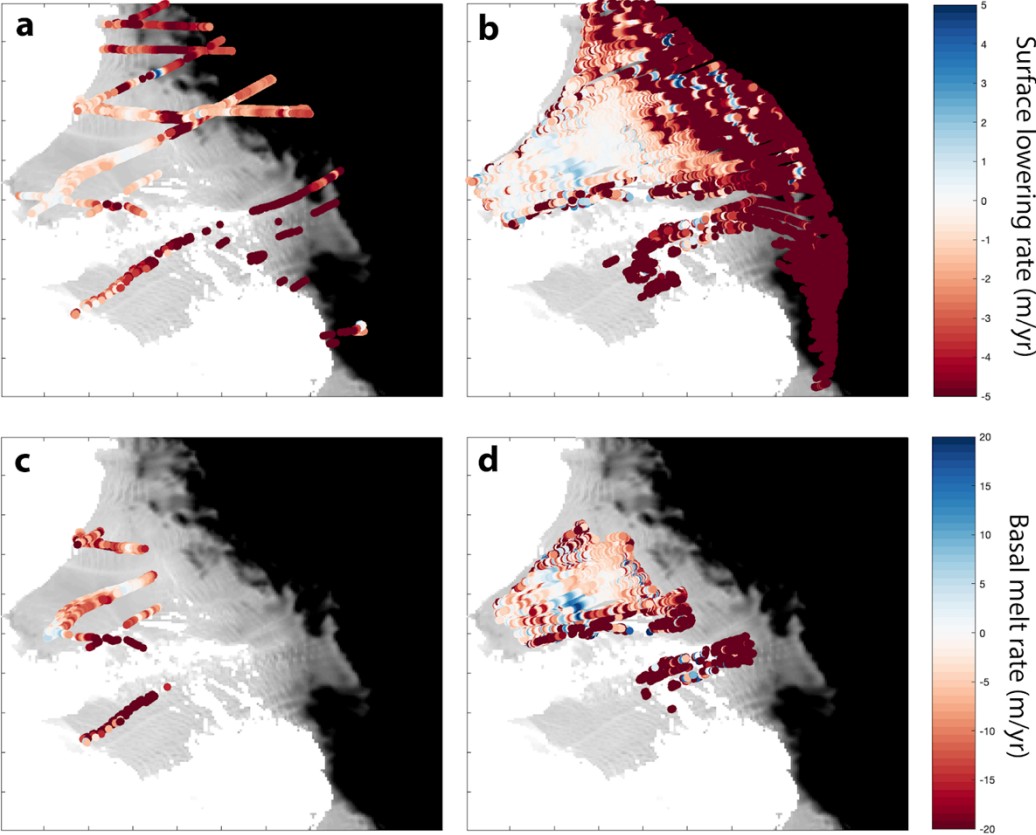

**Figure 7:** Lagrangian surface lowering and basal melt rates. The first row shows surface lowering between ICESat and REMA (panel a) and between REMA and ICESat-2 (panel b). The second row shows basal melt rates between ICESat and REMA (panel c) and between REMA and ICESat-2 (panel d).



The largest rates of surface-height change are found on grounded ice, both due to rapid dynamic thinning in these areas and because surface elevation changes on grounded ice are not hydrostatically compensated. Surface height change on the floating ice shelf is overall much slower. We note an area in the middle of the TEIS that shows relatively rapid surface lowering in panel a, in the same location as relatively rapid surface height increase in panel b. These same areas display rapid melt in panel c and rapid freeze-on in panel d. This small, anomalous region

coincides with a seam between REMA DEM strips, with considerable feathering apparent in the mosaicking. The opposite signs of the signal in the two time periods suggest that the high rates of change here are due to REMA showing incorrectly low surface heights in this area, and not to a real surface height change. However, the consistency in trends in the rest of the data between the two time periods give us confidence in the calculated trends over the majority of the TEIS.


Basal melt rates generally reflect the same patterns as surface-height changes, suggesting that basal melt is the primary cause of surface-height changes on the floating TEIS. Variability in basal melt is particularly high in the shear zone upstream of the pinning point and in the heavily rifted area downstream of the grounding line in the grid-north quadrant. This variability may reflect inaccurate Lagrangian migration of points; areas with extensive

rifting have widely varying ice thicknesses, which would clearly show inaccuracies in tracking of ice parcels. However, we also note that melt rates are typically much higher on near-vertical faces, and these vertical faces migrate laterally as a result, increasing basal melt rate variability in areas with highly variable ice thicknesses, (e.g. Dutrieux et al., 2014).

**4. Discussion**

**4.1 Influence of the Thwaites Western Ice Tongue**

The clearest dynamic control on the TEIS during the first half of the 20-year velocity record presented here is the Thwaites Western Ice Tongue (TWIT). The TWIT is the floating extension of the main trunk of Thwaites Glacier, and has speeds that are typically two to four times higher than those found on the TEIS. A prominent shear margin

separates the TEIS and TWIT, which has had highly variable coherence throughout the record. Early in the record, a relatively short but strong shear margin was present near the grounding line, as indicated with a black arrow in the 2002 MODIS image in Figure 4. The extent of this coherent, strong shear margin increased over the next several years, achieving its greatest length around 2006, as shown in the 2006 MODIS image in Figure 4.



We deduce that this shear margin was strong based on both the lack of large fractures at this time and on the acceleration of the TEIS, supporting the interpretations of other authors (Miles et al., 2020; Mouginot et al., 2014). As shown in Figures 2-4, the TEIS experienced significant acceleration early in the record, peaking around 2005-2007. The 2001-2002 map of speed in Figure 4 shows that the highest speeds on TEIS at this time were found near the shear margin. We interpret this to be a result of large shearing stresses and higher TWIT speeds that dragged

this part of TEIS forwards. This effect became most pronounced during the 2005-2006 season, when the zone of high speeds spread through the middle of the ice shelf, as marked with a dotted oval in the 2005-2006 speed image in Figure 4. By 2007, large rifts developed across the shear margin (see MODIS images in Supplementary Videos 1-5), and by the 2008-2009 season a full separation between the TEIS and TWIT had developed in the shear margin. As it was no longer being dragged forwards by the TWIT, the TEIS decelerated significantly at this point.

The TWIT nearly completely detached and disintegrated in the following years.

Acceleration on the TEIS while the shear margin was strong also added a new set of surface features to the TEIS. A swarm of crevasses opened along the grounding line during this increase in velocity, shown within the dotted oval in the 2006 MODIS image in Figure 4, with more forming at the grounding line over the next few years.

These crevasses are also visible in the Landsat time series of the TEIS shown in Figure 8, starting in the 2005 image where we have marked their formation area with a dotted oval. The crevasse swarm can be seen to advect into the main floating ice shelf throughout the rest of the images in Figure 8; we have indicated this swarm with another dotted oval in the 2020 image.

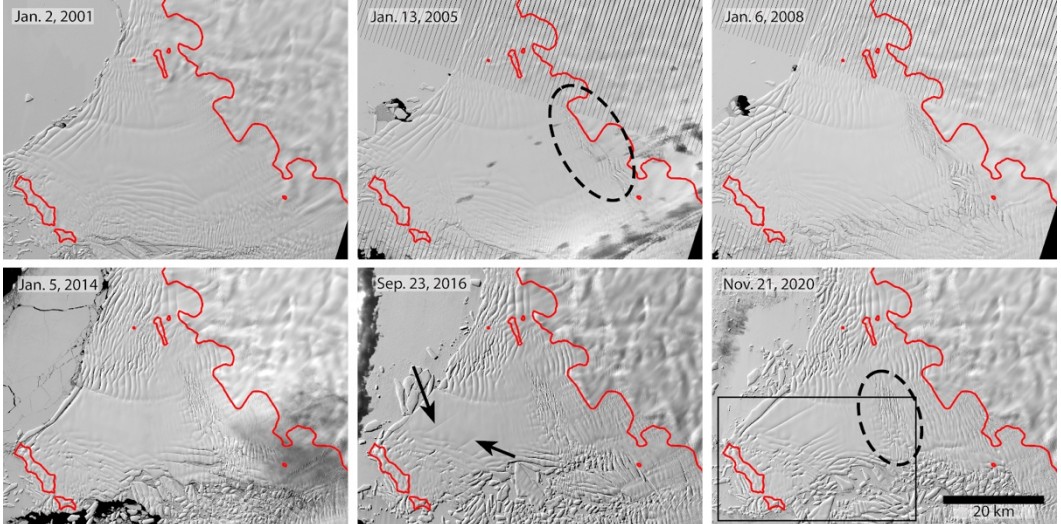

**Figure 8:** Landsat-7 and Landsat-8 time series of the TEIS from 2001-2020. Arrows and ovals are discussed in the text. The rectangle in the 2020 images represents the subset area shown in Figure 9.



## 4.2 The TEIS pinning point and pinning point shear zone

Aside from the influence of the TWIT, the pinning point that confines the TEIS has had the greatest impact on the shelf's spatial patterns of ice-flow speed, direction, and strain rates. This pinning point transmits backstress upstream, as evidenced by the zone of slow velocities consistently found just upstream of the pinning point. That backstress is particularly evident in the 2005-2006 longitudinal strain-rate image in Figure 4, which shows a large zone of negative (compressional) longitudinal strain rates upstream of the pinning point, which is marked with a dotted oval. As the TWIT dragged the TEIS forward at this time, the pinning point provided widespread resistance to this dragging.

Although the backstress transmitted from the pinning point is an overall stabilizing force for the TEIS, the clearest signs of destabilization are now concentrated in this area. As shown in Figure 2, average flow directions on the shelf have rotated toward grid-south (counter-clockwise) during the latter part of the 20-year record. When the TWIT was intact, the presence of the coherent ice tongue largely prevented the ice of the TEIS from outflowing in that direction. With the TWIT removed, TEIS ice flow is now showing strong patterns of divergence around the pinning point, as seen in the 2018-2019 flow direction image in Figure 4 (marked with a dashed oval). This directional divide is also clearly identifiable in the shear-strain values in the 2018-2019 panel in Figure 4 (marked with a dashed oval), which shows a zone of left-lateral shearing that has developed to the grid south of the pinning point and right-lateral shearing to grid north.

Figure 9 provides a close-up of the TEIS pinning point shear zone. The first column shows the 2009-2010 flow direction field, before the flow divergence was distinct, and the 2019-2020 flow direction field, which shows that the pattern has developed into distinct regions of contrasting flow direction with a boundary that closely coincides with the pinning point shear zone. The second column shows longitudinal and shear strain rates derived from Sentinel-1 data during summer 2018-2019. The top panel in this column shows the distinct bands of concentrated longitudinal strain rates noted in the Hovmöller diagram in Figure 6. These bands of concentrated strain likely stretch farther towards the main calving front to grid northwest, but that region is subject to consistent data gaps during the record. Because the strain appears to be concentrating along rifts, shown in the Landsat-8 images in the third column of Figure 9, which extend across most of the shelf in the shear zone, it is reasonable to assume that these concentrated bands of strain also extend across most of the shelf. The 2018-2019 Sentinel-1 shear strain rates in Figure 9 show strain concentration in the same bands but reveal a contrasting sense of shear consistent with the split flow directions.

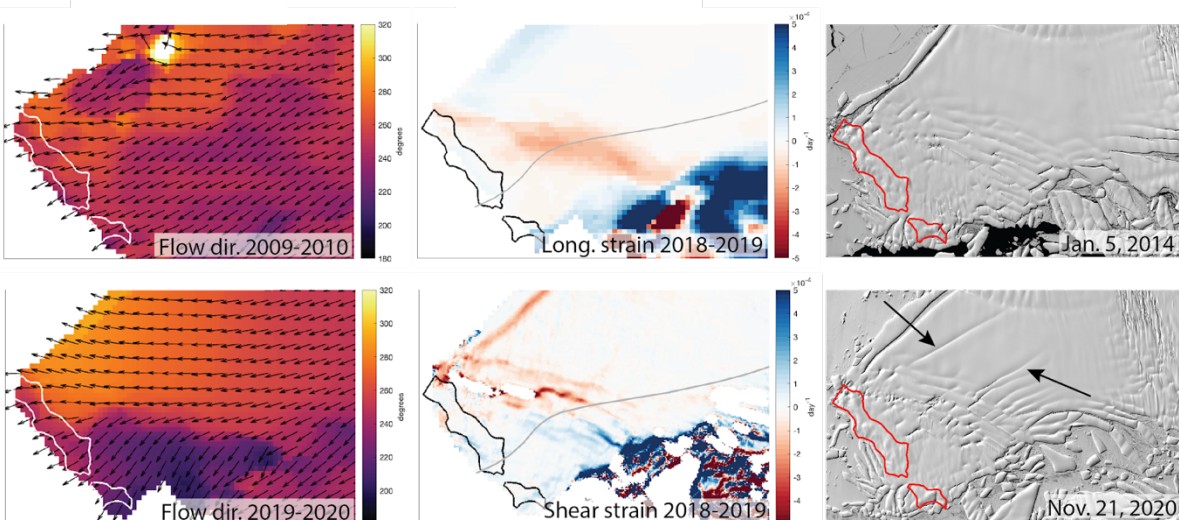

**Figure 9:** Detail of the shear zone upstream of the TEIS pinning point showing the evolution of key features over time. First column: Two-year averages of flow direction, showing a split in flow direction around the pinning point that had not clearly developed by the 2009-2010 image, but had become very distinct by 2019-2020 image. Second column: Longitudinal and shear strain rates derived from Sentinel-1 imagery showing distinct bands of high strain rates in the pinning point shear zone. Gray line is Flowline B (Figure 1). Third column: Landsat-8 images showing the development of large rifts and smaller fractures nucleating in the pinning point shear zone. Red outline in all panels is the grounded pinning point from the 2011 MEaSURES dataset (Rignot et al., 2016). Subset region is indicated in Figure 8.

The Hovmöller diagrams in Figure 6 show that the bands of concentrated strain rates migrate towards the pinning point over time, as does the region of higher speeds upstream of the pinning point shear zone. This migration is

475  occurring at approximately the same speed as ice flow, which may indicate that these dynamic changes are advecting with the ice as the TEIS continues to adjust to the loss of the TWIT. However, the migration of concentrated strain and higher velocities towards the pinning point may alternatively or additionally indicate that the TEIS pinning point is ungrounding, removing backstress that has prevented this change in the past. The new flow divergence around the pinning point suggests that thinner ice is being delivered to the pinning point, which

480  could promote ungrounding. Analysis of pinning point evolution is ongoing and will be presented in a separate paper.

Simultaneous with the development of divergence in ice flow around the pinning point, new, relatively small rifts have begun to form within the shear zone, and large, laterally extensive rifts have nucleated from the shear zone





485    and extended into the middle of the shelf. These large rifts first appear in 2016 and are marked in the 2016 Landsat-
8 image in Figure 8, as well as in the 2020 Landsat-8 image in Figure 9. As these rifts have formed within regions
of high shear strain in the pinning point shear zone, they are likely caused at least in part by the new pattern of
flow divergence around the pinning point.

In addition, Figure 7 shows sustained, concentrated areas of relatively high rates of surface lowering and basal
melt in the pinning point shear zone. Surface lowering and basal melt rates in this region are highly spatially
variable, which may be related to variability in ice thickness and basal slope due to the presence of rifts and basal
crevasses. Large differences in ice thickness may exaggerate errors in the Lagrangian migration of points, resulting
in false variability in surface height change and basal melt. However, there is also reason to believe that basal melt
rates should be highly variable in fractured basal ice, as cold meltwater insulates relatively horizontal ice from
melt while melt rates can be much higher on ice faces that are closer to vertical (e.g. Dutrieux et al., 2014). These
values may therefore reflect localized high rates of real basal melt and thinning in this shear zone, which may also
have contributed to the formation of large rifts within the shear zone.

**4.3 TEIS and ocean forcing**

Figure 7 shows that patterns of surface elevation change and patterns of basal melt on the floating ice shelf are
very similar. Basal melt was calculated from mass conservation, taking into account surface mass balance and ice
thickness divergence to explain the observed changes in surface height. Similarity between the patterns of basal
melt and surface lowering suggests that surface mass balance and ice flux divergence contribute little to surface-
height changes, and that the vertical TEIS changes are driven by ocean forcing.

This is not an especially surprising result, as many studies (e.g. Pritchard et al., 2012) have shown that dynamic
changes in the Amundsen Sea are driven by strong basal melt forcing by warm Circumpolar Deep Water (CDW).
However, although CDW presence leads to an overall increase in ice shelf basal melt and thinning, spatial and
temporal details may be much more complex. Seroussi et al. (2017) ran a 50-year simulation of basal melt beneath
the TEIS, showing that melt rates initially decrease as the ice shelf base thins out of the reach of CDW, before
melt rates increase with continued climate forcing. This separation from CDW on the relatively flat basal
topography in the mid-TEIS may be responsible for the relatively low basal melt and thinning rates in this area.
Although the ice draft is not significantly different than the middle of the shelf, the higher basal melt and thinning
rates seen near the grounding line and pinning point shear zone may be due to the presence of steep basal
topography prone to faster melt.



Another potential control on thinning in these areas may be directly related to patterns of ocean currents beneath the TEIS. Wåhlin et al. (2021) used CTD casts and an autonomous underwater vehicle to measure water properties

near and beneath the TEIS during a 2019 cruise. Identified pathways of warm water inflow include previously underestimated branches from the east, roughly following bathymetric troughs beneath the main calving front of the TEIS, and significant heat inflow through troughs from the north along the TWIT/TEIS shear margin. These observational data add considerable detail and new information to results produced by Nakayama et al. (2019), who used a high-resolution ocean model to show that increased basal melt rates on the TEIS coincide with faster

sub-ice-shelf currents. Modelled (Nakayama et al. 2019) and observed (Wåhlin et al. 2021) warm inflows coincide roughly with the areas where we observe relatively large rates of thinning and bottom melting (Figure 7), including near the grounding line to the east and in the shear zone upstream of the TEIS pinning point.

These results suggest that direct ocean forcing is a possible explanation for the earlier unpinning and disintegration

of the TWIT. Modelled currents and melt rates were found to be faster beneath the TWIT than the TEIS (Nakayama et al., 2019), and the heat transport in one of the deep troughs leading under the TWIT/TEIS shear margin was very high (Wåhlin et al. 2021). Furthermore, Wåhlin et al. (2021) suggest that the ocean heat transport observed to be currently influencing the TEIS pinning point is unsustainably high and may lead to unpinning and destabilization in the style of the TWIT. Assuming that the TEIS pinning point experienced stable melt rates in

previous decades, the observed high heat fluxes may be due to an externally forced change in ocean circulation, and/or could relate to a positive feedback where a reduction in ice-shelf draft due to basal melt allows increased inflow of warm water. So, while the observed TEIS ice flow changes may be responding to ice-dynamic controls from the TWIT and upstream ice, they may also be directly due to ocean circulation changes that have increased heat fluxes and basal melt, thinning and weakening the ice shelf near the crucial TEIS pinning point.


## 5. Conclusions and future outlook for the TEIS

The past 20 years of change on the TEIS were dominated by dynamic interaction with the neighbouring TWIT. Early in the record (~2000-2006), the TEIS experienced large lateral stresses from the more rapidly flowing TWIT, causing the TEIS to accelerate. This was followed by rapid TEIS deceleration as the TWIT/TEIS shear margin

weakened and the TWIT decoupled and disintegrated around 2007. The TEIS then developed new, independent flow patterns, including an overall ice velocity increase. The pinning point responsible for maintaining TEIS stability has now become an epicentre of destabilization. During the last several years of the record, ice flow has strongly diverged around the pinning point and strain rates have concentrated in narrow bands in the shear zone upstream. Simultaneously, significant fracturing has nucleated within the region of high strain rates and several

rifts have penetrated much of the TEIS's central region.



Sparse measurements of surface lowering rates are available between ~2003 and 2014 from ICESat and REMA, with much more detail available between ~2014 and 2020 from REMA and ICESat-2. These data show generally low thinning and basal melt rates in the central TEIS, with much more variable and overall higher basal melt rates

near the grounding line and in the shear zone upstream of the pinning point. The presence of relatively high thinning rates is particularly important in the pinning point shear zone, where basal melt may be partially responsible for weakening that has led to new rift formation.

Both the vertical and horizontal changes observed on the TEIS over the last 20 years indicate progressive

weakening and destabilization of the floating ice shelf. There is no indication that these trends will reverse in the future. Increased forcing by CDW is likely to continue (e.g. Holland et al., 2019), and upstream acceleration and thinning of Thwaites Glacier means that ice advected onto the shelf may be more damaged (e.g. MacGregor et al., 2012). The patterns of dynamic instability that we have observed indicate that weakening will enhance over time (see also Joughin et al., 2014; Rignot et al., 2014). Based on this analysis, the future of the TEIS looks much like

what we have already seen on the TWIT: a total or near-total loss of the floating ice shelf, removing the buttressing connection with the pinning point and resulting in acceleration of grounded ice. We suggest that final disintegration of the TEIS will occur in one of three possible ways:

1. The surface crevasse swarm that nucleated at the grounding zone around 2005 will continue to advect,
reaching the central region of the shelf that is now penetrated by large rifts. These damaged areas will join in 10-20 years and may destabilize the TEIS throughout its central region. The impact of this event will depend on whether new, large rifts continue to nucleate from the pinning point shear zone, and the evolution of the crevasse swarm (further extension, or healing) as it continues to advect. The condition of the crevasse swarm will depend largely on mid-shelf longitudinal strain rates, which are primarily
compressional, but are trending towards neutral or extensional.

2. The ice shelf may decouple from the pinning point due to large-scale failure in the pinning point shear zone. Based on the rapid development of rifting within the shear zone in the last ~5 years, this could plausibly occur on a timescale of years to decades. The pace of this failure is likely to be set by the basal melt rate and the continued concentration of stress along large rifts extending across the pinning point
shear zone. We note, however, that break-up of other ice shelves has been highly non-linear and that a sufficiently thin and weak shelf can break up very rapidly.

3. Continued ocean-forced thinning of the ice shelf and advection of thinner ice onto the pinning point will result in partial or complete unpinning of the ice shelf and loss of integrity. The extensive flow changes



and migration of high velocities towards the pinning point over the last decade suggest that this process
is underway and could destabilize the shelf in one to two decades.

**Acknowledgments**

This work is from the TARSAN project, a component of the International Thwaites Glacier Collaboration (ITGC).
Support from National Science Foundation (NSF: Grant 1929991) and Natural Environment Research Council
(NERC: Grant NE/S006419/1). Logistics provided by NSF-U.S. Antarctic Program and NERC-British Antarctic
Survey. Sentinel-1 data were provided by the Copernicus Program of the European Commission. We thank
Fernando Paolo, Luc Girod, and Richard Alley for advice on data preparation methods, and Anna Wåhlin for her
comments on this manuscript.

**Author contributions**

K. Alley led data analysis and writing. C.T. Wild produced alpha maps and assisted in analysis of altimetry and
REMA data. A. Luckman analysed all Sentinel-2 data and provided figures for the manuscript. C.T Wild, T.A.
Scambos, M. Truffer, E.C. Pettit, and A. Muto assisted in data analysis and manuscript planning. B. Wallin and
M. Klinger provided data and expertise in MODIS imagery and PyCorr processing. T. Sutterley and C. Hulen
assisted in processing of ICESat-2 data. S. F. Child assisted in analysis of the REMA DEM in relation to ICESat
and ICESat-2 data. J.T.M. Lenaerts, M. Maclennan, E. Keenan, and D. Dunmire provided data processing for firn-
air thickness and surface mass balance used in basal melt rate calculations. All authors participated in the writing
and revision of the manuscript.

**Competing interests**

The authors declare no competing interests.

**Data availability**

Data sources are cited in the text. Derived velocity and strain-rate records are available through the US Antarctic
Program Data Center (Alley et al. 2021).

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
