# Peer review of "Two decades of dynamic change and progressive destabilization on the Thwaites Eastern Ice Shelf"

_The Cryosphere, 2021_

## Author Comment (AC2)

**Section 1: Basal melt rate error analysis**

First we find the error associated with ice thickness. We rearrange equation 1 from the text to solve for H, and then propagate the error:

$$Z_s = \left(1 - \frac{\rho_i}{\rho_w}\right) H + \left(\frac{\rho_i}{\rho_w}\right) h_a$$

$$\frac{Z_s - \left(\frac{\rho_i}{\rho_w}\right) h_a}{\left(1 - \frac{\rho_i}{\rho_w}\right)} = H$$

Error in first $(Z_s)$ term:
 REMA error in $Z_s$: $\sigma_{1REMA} = 6\ m$
ICESat/2 error in $Z_s$: $\sigma_{1ICES} = 0.2\ m$

Error in second $(h_a)$ term:
Error in $h_a$: $\sigma_{h_a} = 1\ m$

$$\sigma_2 = \left(\frac{\rho_i}{\rho_w}\right) \sigma_{h_a} = 0.89\ m$$

Error propagation for addition and subtraction:

$$\sigma_H = \sqrt{\sigma_1^2 + \sigma_2^2}$$

$\sigma_H = 6.1\ m$ for REMA and 0.91 m for ICESat/ICESat-2

Then divide by the constant value on the bottom:
$$\sigma_{HREMA} = 57\ m$$
$$\sigma_{HICES} = 8.6\ m$$

$$\frac{DH}{Dt} + H(\dot{\epsilon}_{trans} + \dot{\epsilon}_{lon}) = \dot{m}_s + \dot{m}_b$$

Error in first term:
Start with error propagation for addition and subtraction, then divide by Dt:
$$\sigma_{DHDT} = \frac{\sqrt{\sigma_{HREMA}^2 + \sigma_{HICES}^2}}{DT}$$

$\sigma_{DHDT} = 11.5$ m/yr for REMA to ICESat-2
$\sigma_{DHDT} = 7.2$ m/yr for ICESat to REMA

Error in second term:

$$\sigma_2 = 250 * \sqrt{(4*10^{-4})^2 + (4*10^{-4})^2} = 0.14 \, m/yr$$

Error in third term: 0.1 m of ice equivalent/yr

Error in basal melt rate:

$$\sigma_{\dot{m}_b} = \sqrt{\sigma_{DHDT}^2 + \sigma_2^2 + \sigma_{SMB}^2}$$

$\sigma_{\dot{m}_b} = 11.5$ m/yr for REMA to ICESat-2
$\sigma_{\dot{m}_b} = 7.2$ m/yr for ICESat to REMA

**Section 2: Revision of Figure 7**

---

## Author Response (AR1)

**Two decades of dynamic change and progressive destabilization on the Thwaites Eastern Ice Shelf**

**List of changes in the manuscript:**

Note: Line numbers are slightly different between the track changes and clean versions. We've noted both sets of line numbers, indicated by: T: Track-changes line numbers, and C: Clean copy line numbers

Lines T 111-126, C 111-126: Additional information on datasets used and datasets that could not be used in our velocity analysis.

Lines T 160-161, C 152-153: Information on tidal corrections in the Sentinel-1 analysis that was accidentally omitted in our first submission.

Lines T 172-175, C 165-167: Clarification in the text about the true error in our stacked velocity products.

Lines T 184, 275, and 287, C 176, 266, and 279: Added equation numbers to the manuscript

Lines T 187-188, C 179-180: Additional clarification on error in velocity products.

Lines T 277-284, C 267-274: Additional explanation on error calculation in firn-air content measurements.

Lines T 291-294, C 282-285: Corrections to our error analysis, reference to detailed equations in Appendix A, and context for basal melt rate error estimates.

Lines T 415, C 395: Revised Figure 7 and figure caption.

Line T 626, C 610: Correction of typo (Sentinel-1 rather than Sentinel-2).

Lines T 747-749 and 755-757, C 730-733 and 737-740: Additional citations

Lines T 840-880, C 829-869: The addition of Appendix A, which describes our error propagation for error estimates in basal melt rates.

**Responses to reviewer comments**

*Reviewer comments in Italics;* Author responses in normal font

**Reviewer 1:**

**Reviewer comment:** *This manuscript by Alley and others reports on the evolution of the Thwaites Eastern Ice Shelf over the past 20 years. They use an impressive suite of remotely sensed observations to show its transition from a flow regime largely controlled by the faster flowing Thwaites Ice Tongue, to a largely independent regime following the near complete loss and structural weakening of the Thwaites Ice Tongue around 2007. Importantly they also show the evolution of the TEIS after its decoupling with the Thwaites Ice Tongue and show a progressive weakening in the shear zone near its pinning point and a clear divergence in flow, which is essentially ripping the ice shelf apart. On the basis of these observations the authors hypothesize three potential routes to further destabilization of the TEIS over the timescale of years to a couple of decades. These are important conclusions given the importance of the TEIS in providing buttressing to the Thwaites Glacier.*

*This is a very impressive manuscript and I found it an enjoyable read. The manuscript is packed with excellent detail, it is well illustrated and I found it very easy to follow. The discussion and conclusions are appropriate given the results presented. I recommend publication as is.*

**Author response:** We sincerely thank the reviewer for taking the time to read our manuscript and for the very positive comments. We have done our best to maintain the current methods and conclusions endorsed in this review as we responded to the comments from Reviewer 2.

**Reviewer 2:**

**Reviewer comment:** *Alley et al. present a manuscript describing the evolution of the Thwaites Eastern Ice Shelf (TEIS). As Thwaites is a key glacier for understanding and predicting the future contribution of the West Antarctic Ice Sheet, this study of TEIS brings new information on its dynamic and geometric changes that are certainly important to the community. TEIS buttresses a large portion of Thwaites that has displayed only moderate dynamic changes compared to the main ice tongue. Losing this remaining "barrier" could mean a larger Thwaites debacle in the future and thus an increased contribution of the glacier to sea level rise. To study TEIS, the authors used remotely-sensed observations from MODIS, Landsat-8, Sentinel-1 to document its dynamic and complement them with elevation measurements from ICESAT-1&-2 and optical stereo elevation models from REMA to derive lagrangian elevation changes and basal melt.*

*If the approach is globally sound, I regret however that this study does not use the best existing data or methodology, and that the presentation of some results or the calculation of errors are not more careful because it may weaken the credibility of the results. Therefore I suggest a major revision of the manuscript before it is suitable for publication.*

**Author response:** We thank the reviewer for spending the time to carefully read and consider our manuscript, and for endorsing our sound approach. We believe that, with further explanation, the reviewer will be pleased with our thoroughness in our current analysis, and agree that we have used the best available data for our specific aims. In the text below, the reviewer has identified one important method that we forgot to mention in the text, made an excellent suggestion for improving one of our figures, and requested further clarification about our error analysis methods. We offer improvements based on all of these comments. However, the reviewer's main comments also suggest adding 10 additional or extended datasets to our manuscript and focusing on alternative research goals. We sincerely hope that the reviewer or others will use many of these datasets in the future to carry out the suggested analyses, which are highly complementary to the work presented in this study. However, they are beyond the scope of this manuscript and would not alter our conclusions, which the reviewer agrees are important to the continued study of Thwaites Glacier.

*Reviewer comment:*

*Speed observations:*

*While their results are interesting in documenting the progressive weakening of the floating ice shelf, I believe that the existing observations to analyze the TEIS dynamic evolution are underutilized. A large part of the analysis is based on the use of MODIS with a quite low spatial resolution. These data are used to calculate velocity changes, strain rate evolution but also to calculate elevation changes and submarine melt with a Lagrangian approach. However, one may wonder about the robustness of these calculations in view of the large errors associated with these measurements. With an error of several hundred meters per year for an ice shelf flowing at less than 1 km/year, the error on the flow direction is quite large (several tens of degrees). We can therefore question the validity of the measurements with the Lagrangian approach, as well as the calculations of strain rates. Even with filtering and large spatial smoothing, it is clear that the MODIS results form unrealistic patches where the flow direction and amplitude do not seem very homogeneous as it is visible in Figure 9 top-left. This is also visible in Figure 6 where the combined registration seems to bring many biases (especially in 2018) that are not present in the Sentinel-1 record alone. If the MODIS observations were the only ones available to document the velocities in the years 2000 to 2010, I would not see too many problems to use them as they would be the only existing source of information. But, as far as I know, there are many other instruments that allow measurements during this period (even if, of course, this would not match the amount of observations obtained during the last years with the Sentinels and Landsat). Thus the authors could have used higher resolution data from ENVISAT/ASAR, ALOS/PALSAR, RADARSAT, Landsat-7 (between 1999 and 2003) or even ASTER which are publicly available. Some of the speed measurements using these instruments are already available at NSIDC if I am not mistaken and should therefore be considered.*

**Author response:** We explored the datasets suggested here during the initial preparation of our manuscript, and unfortunately they do not provide the coverage or accuracy that the reviewer is hoping for here. Data from ENVISAT/ASAR, ALOS/PALSAR, and RADARSAT, along with several other sensors, are incorporated into the MEaSUREs velocity data that are distributed through NSIDC. Annual velocity grids are available starting in 2005, five years after the beginning of our analysis. Our understanding of the controls on Thwaites Eastern Ice Shelf (TEIS) flow depend crucially on the time period between 2000 and 2005, when the influence of the Thwaites Western Ice Tongue (TWIT) evolved very rapidly. Without data from that time period, we would be missing significant evidence for our conclusions, and

these data are provided primarily by MODIS. Furthermore, the annual grids that are available from MEaSUREs lack the spatial resolution and coverage provided by the MODIS data. Annual MEaSUREs grids are provided at 1-km resolution, while our analysis is at 500-m resolution, and several of the grids have significant missing data in the central TEIS. While these data could be processed at a higher resolution, the required work would be appropriate for a separate project, and reprocessing would regardless not solve the coverage issue in either space or time. Ultimately, the available InSAR data would not improve the data needed for our conclusions, and the inclusion of available products would decrease our spatial resolution, which is important for accurate Lagrangian analyses later in the paper. We have noted this in lines T 117-119, C 117-119 in the text.

The reviewer also suggested using Landsat-7 between 1999 and 2003 and ASTER. We have already included all available Landsat-7 data in our analysis, including from that time period, as it is included in the ITS_LIVE data cited in the text. We have clarified this in lines 111-115 in the text.. We worked extensively with ASTER during our data preparation, and unfortunately found that it was not suitable for the analysis. There are relatively few images of this area available from ASTER, and many of the ones that are available suffer from cloud cover. During the 13-year period between 2001 and 2013, when Landsat-8 data are unavailable, three seasons lack any cloud-free imagery of the TEIS at all, and four more have a single day of data with incomplete coverage of the ice shelf, severely limiting the potential for successful velocity correlations. In addition, many mid-shelf correlations from ASTER imagery are unsuccessful. This is now noted in lines T 115-117, C 115-117.

Aside from the lack of availability from other datasets, we find that the MODIS data are sufficiently accurate for our analysis. The error figure that the reviewer cites of "several hundred meters per year" and "several tens of degrees" could be reasonable for a single correlation, but it overestimates the error for the averaged grids we have provided; as shown in figure 2, error bars are at maximum approximately +/-100 m/year, typically under 10% of the flow speed, or +/-10° (these error bars are considerably smaller later in the record, when Landsat-8 data are available). We have revised the text in lines T 172-175, C 165-167 to make the true error ranges in our stacked velocity grids clearer. Furthermore, the velocity changes that we discuss in the conclusions that are important in understanding the overall ice-flow history on the TEIS are well outside the error bars, giving us confidence in the conclusions.

Overall, we believe that we have used all available velocity datasets that add value to this analysis, and that the data that are available are sufficient for the conclusions we have drawn.

**Reviewer comment:** *Concerning the Sentinel-1 processing, it seems that the tidal signal is not corrected while this signal strongly affects the range component of the Sentinel-1 at 6 and 12-day repeat cycles. This problem is probably mitigated by the fact that the data are averaged by quarters. Nevertheless, this may lead to additional errors that are currently not taken into account and therefore should be at least discussed to evaluate the impact it has on the data.*

**Author response:** Thank you for catching our omission - we have corrected for tides using CATS 2008 in the Sentinel-1 processing, we just forgot to note this correction in the text. We have noted this in lines T 160-161, C 152-153.

**Reviewer comment:** *Elevation data and basal melt rate:*

*Regarding the elevation data, the authors use the REMA as a reference to compute lagrangian elevation changes compared to IceSAT (2002-2009) and IceSAT-2 (2018-present). REMA is vertically referenced to CryoSAT-2 elevation. Why not use directly the CryoSAT-2 observations ? If this is due to possible error due to penetration in Ku-Band in firn and/or snow, then the same concern could be raised for the calibration of REMA.*

**Author response:** Many other authors (e.g. Smith et al. 2020) have carried out Eulerian analyses of ice-thickness change and basal melt on Antarctic ice shelves, including the TEIS. These analyses are well-suited to large-area averages, as effects of ice-thickness advection largely cancel out. However, we wanted to examine spatial variability in ice- thickness change and basal melt at a higher resolution than has been necessary for many past analyses, which requires a Lagrangian approach. Because Lagrangian approaches require migration of measurements from the measurement epoch to a reference grid, we either need to use a full-coverage DEM or to interpolate between points from an altimeter. As the REMA mosaic shows, the topography of the TEIS is complex and varies on spatial scales smaller than can be captured by interpolating between CryoSat-2 point measurements. It is therefore better to use the altimetry data to reference a full-coverage DEM, as has been done with REMA.

*Reviewer comment: It is also unclear if the authors have used individual REMA strips from GeoEye and Worldview acquired between in 2013 and 2014 and then referenced them to CryoSAT-2 themself, or if they used an already mosaicked REMA product where they have no real control on the quality of the results. I imagine that it is the latter because otherwise there would have been the possibility to correct for the tides which apparently was not done. Here several other questions are raised: (1) why not use the complete REMA archive which provides data over a longer period (2012 to 2018) than 2013-2014? It would be possible to calculate the displacement directly on the REMA DEMs which would allow to obtain almost perfect co-registration for the Lagrangian calculation (much better than using flow velocities obtained by other sensors). Obviously the vertical errors would remain high (+/- 6m) but that does not seem to be too much of an issue here.*

**Author response:** While some REMA DEMs are available spanning the mentioned time period between 2012 and 2018, almost all the coverage in this area is available between 2014 and 2016. Even with the high rates of change on TEIS, the errors associated with differencing two REMA DEMs during this time period would be too high to obtain meaningful results. Two other barriers stand in the way of this sort of analysis: 1. Lagrangian analysis requires a reference grid with complete or near-complete coverage, so that there is data availability at any location a point migrates to. This would necessitate mosaicking available DEM strips, which is exactly what has been done with REMA; we have neither the computing power nor the expertise to do this mosaicking better than the original REMA authors, which is why we have used a single REMA mosaic tile. 2. We have shown that velocities on the shelf have changed significantly over time. With incomplete coverage from REMA strips, we would not be able to obtain annual velocity grids that capture these changes, instead having to rely on longer time-averages that would miss these changes and introduce larger errors into the Lagrangian analysis. Our annual velocity analysis is thus more suitable for Lagrangian calculations on the TEIS.

*Reviewer comment: (2) Why not use CryoSAT-2 directly, using these observations, there would also be the possibility to correct the tides which cannot be done in the REMA mosaic.*

**Author response:** As noted above, a Lagrangian analysis requires a gridded dataset, and gridding of available CryoSat-2 data does not have the necessary resolution to capture the high spatial variability in TEIS topography.

*Reviewer comment: (3) As a complement, there might have been the possibility to obtain high resolution elevation data from TanDEM-X that would have been the perfect complement for this study.*

**Author response:** As the reviewer notes, this is a great idea for a complementary study that could extend the work done here. However, it is well outside the scope of our methodology, and the conclusions we have drawn are well within the error bounds of our data, so this additional dataset is unnecessary in the current study.

*Reviewer comment: (4) Lidar data from Operation IceBridge probably exist during the studied period and would certainly provide constraints from REMA DEMs or add additional measurements to IceSAT. Why not include them ?*

**Author response:** As in the previous question, a complementary study could certainly decide to go in this direction, but it would add little to the analyses we have presented. IceBridge lidar data are sparse on the TEIS; the year with the most extensive data coverage is 2009, when 6 flight lines crossed the TEIS. All other years have even sparser coverage. Coverage that coincides with collection of REMA DEMs is far too sparse for effective vertical referencing. In addition, IceBridge data collection began after the ICESat era, which means that there is little to no separation in time between the available IceBridge transects and REMA DEMs. With less vertical change over a shorter period of time, trends would not fall outside the error range. We believe the IceBridge data are extremely valuable for analyses of specific areas, and our team has a separate study in review that utilizes these data, but they add little to the large-scale analyses that are the subject of this study.

*Reviewer comment: If the authors seem to have done a good job in correcting for tides, taking into account the firn to convert elevation to ice thickness and surface mass balance in the melt rate calculation, it is unfortunate that these corrections are not shown as supplemental material of the paper as maps. In the same way, error maps could be shown to evaluate spatially the robustness of the different observations. I am also unsure if the evolution of firn air content over time is taken into account when calculating thickness changes.*

*The error calculation for the elevation changes and for the melt rate calculation remains also rather unclear. The errors for the firn and for the SMB are not provided. The error for elevation changes are estimated to be to the order of 1 m/yr, therefore the error on melt rate without the additional errors coming from firn, surface mass balance or flux divergence should be alone about 10 times larger (9.41 to be exact with the chosen density in seawater and ice) but surprisingly the authors found basal melt error lower than for the surface elevation changes. This needs to be clarified.*

**Author response:** Tide corrections are derived from the freely available CATS2008 model (https://www.esr.org/research/polar-tide-models/list-of-polar-tide-models/cats2008/), and maps of tidal variation can be readily created using this model. As the data were collected at many different times, it would be impractical to show all of these maps, even in supplementary information. The model used for firn air content and surface mass balance are at a very coarse spatial resolution, so a single average value is available for the TEIS; our error is therefore an area-averaged estimate for the TEIS, and showing this as a map

would be uninformative. We have adjusted our analysis to include the firn-air content (FAC) generated from BedMachine, which takes into account spatial variability across the TEIS, and use the SNOWPACK model to estimate a spatially averaged variability over the time period of our study. This variability in time is used to make an error estimate of 1 m for FAC, which we use in our error analysis for basal melt rates. These adjustments are noted in lines T 277-284, C 267-274.

Thank you for the very detailed reading of our manuscript; we inadvertently used the error associated with surface height change (dh/dt) rather than the error for ice thickness change (dH/dt) in our basal melt error analysis. We have corrected the values in the text in lines T 291-292, C 282-283 and added detailed explanations of our calculations in Appendix A. The correct basal melt error calculations are: 11.5 m/yr for REMA to ICESat-2, and 7.2 m/yr for ICESat to REMA. Note that, despite the high values attributable primarily to the uncertainty in REMA, the areas of high basal melt that we have noted in the text as important (particularly in the shear zone upstream of the pinning point) have basal melt rates in the range of 10-20 m/yr, with the highest values more than 50 m/yr, which is well outside of this error range, and does not call any of our conclusions into question. We also note that the consistency between the ICESat/REMA and REMA/ICESat-2 epochs suggests that error over most of the shelf is considerably lower than this estimate, although the sparse data from ICESat prevents a more robust analysis of this similarity.

*Reviewer comment: Figure 7 is not very appealing. The use of point shapefile to show changes in surface elevation and basal melt makes the graph quite messy and complicated to read. It would have made much more sense to create an interpolated and filtered spatial map from this point cloud. An evaluation of the total melt and a comparison with existing results would have been welcome. Melt rates are evaluated for two periods 2003-2013 and 2013-2020 with IceSAT and IceSAT-2, respectively. However I could not find any analyses of potential changes in melt pattern or elevation changes. How much the basal melt has changed ? What are the implications of relative changes in thickness ?*

**Author response:** We agree with the reviewer that the point representation we have presented is not ideal. We had presented it in this way in order to have a consistent symbology between the ICESat and ICESat-2 data points. While we could create an interpolated and filtered spatial map of the ICESat-2 data with reasonable coverage of the ice shelf, a similar presentation of the ICESat data is not reasonable, as they are far too sparse for interpolation on an ice shelf with so much topographic variability. We have revised Figure 7 with the ICESat data appropriately left in a point representation and the ICESat-2 data gridded across the entire shelf.

Because the ICESat data are so sparse and variability in thinning and basal melt rates so high, in addition to relatively high error estimates, our opportunity for comparison is very limited and we can have very little confidence in generalized statements of regional patterns of change based on the available data. However, consistency between the datasets in a few key areas of high basal melt rates and thinning rates suggests persistent forcing on average over the last two decades. As explored in detail in our discussions, this has important implications for the weakening of already-weak areas of the ice shelf. A more detailed study focusing on melt rates and changes in melt rates would be valuable, but it is not the goal of the current study.

*Reviewer comment: A vertical cross-section along the flowlines would have proved useful to illustrate the melt rate and thickness changes along TEIS, especially close to the pinning point and the grounding line. Potentially this could have been compared with OIB radar*

*flight lines directly measuring thickness at different dates. Overall, I think that the results and discussions about melt rate and thickness changes need to be more quantitative. Indeed, there is a crucial need to better model the interactions between the ocean and the glaciers in this region. By providing a more rigorous and quantitative analysis of melt patterns and evolution, the authors would provide an important input to a better understanding of the circulation of ice shelf cavities in the Amundsen Sea embayment.*

We have another paper in review (Wild et al., TCD) that uses OIB radar flight lines to look at changes near the grounding line and pinning point. While we agree that a more quantitative discussion of change would be useful to the community, we have done what is appropriate for the available data, and an analysis of melt patterns and pinning point evolution is not primary the goal of this paper.

**Reviewer comment:**

*Other specific comments:*

*Figure 1 shows the grounding line evolution from 2004 to 2017. It is again rather unclear why the authors have not used published datasets (NSIDC) that provide grounding line position since 1992. It would have appeared that the delimitation of the grounding of 2004 is not correct. Already in 1996, the InSAR grounding line was several kilometers further back in many places.*

**Author response:** The 2004 grounding line is the published grounding line as downloaded from NSIDC. The citation is provided in-text in the caption (Bindschadler et al. 2011) and as a full citation in the list of references. As our analysis begins in 2000, a 1996 grounding line would be less relevant to our paper.

**Reviewer comment:** *l597: The authors mentioned that Adrian Luckman analyzed "Sentinel-2", I believe that the authors meant Sentinel-1, as no mention of Sentinel-2 is done in the manuscript.*

**Author response:** Thank you; we have corrected that typo in line T 626, C 610.

**Reviewer comment:** *The authors provided datasets used in the study at the following link: https://doi.org/10.15784/601433. This is a very good initiative and I hope that if the manuscript is accepted the link will work successfully as it is not currently the case.*

**Author response:** We echo the reviewer's emphasis on the importance of sharing data. The link works just fine for us; we hope the reviewer will contact the USAP-DAC (https://www.usap-dc.org/contact) to address any technical problems they are facing.

---

## Author Response (AR2)

**Two decades of dynamic change and progressive destabilization on the Thwaites Eastern Ice Shelf**

**List of changes in the manuscript:**

Note: Line numbers refer to the track-changes version of the manuscript.

**Changes to figures:**

Figure 1: Added 2000 grounding line, adjusted box sizes to 5 km x 5 km and updated location of Site 1, updated caption accordingly.

Figure 2: Updated dataset

Figure 3: Updated dataset, increased text size

Figure 4: Updated dataset, increased text size, added grounding lines to MODIS images, added labels for ovals, updated caption

Figure 5: Updated dataset, increased text size

Figure 6: Updated dataset, increased text size, clarified caption

Figure 7: Updated interpolation and color bar label, updated caption

Figure 8: Updated caption

Figure 9: Updated dataset

**Changes to the text:**

100-101: Added text emphasizing comparison of short-term combined record to Sentinel-1 only record

113-118: Added text describing the masking of MODIS data above the grounding line

129-131: Added citations demonstrating the reliability of MODIS data for velocity records

144-145: Described additional filtering for the long-term velocity record

234-237: Clarified use of ASAID grounding line product

294-297: Clarified terms of the mass conservation equation used to calculate basal melt

Section 3.1 (305-403): Adjusted text to properly describe observations of averages within boxes, now that they are larger and the grounding zone site has been moved. Removed mentions of inaccuracies in MODIS data above the grounding line, as these data have now been removed from the record.

Section 3.2 (403-440): Clarified description in text of difference between our short-term combined record and the Sentinel-1-only record

454-455: Noted interpolation method

464-466: Clarified possible reasons for the surprising mid-shelf freeze-on signal

469: Clarified inclusion of ice thinning/thickening in calculations

494, 509, 524, 532, 534: Added reference to numbered dashed ovals in Figure 4

584: Clarified inclusion of ice thinning/thickening in calculations

596: Clarified possible reasons for the surprising mid-shelf freeze-on signal

References: Added citations

**Response to reviewer comments**

*Reviewer comments in italics;* Author responses in normal font

**Reviewer 2:**

***Reviewer comment:*** *Alley et al. have addressed or corrected some of my comments regarding data processing and error analysis.*

*Nevertheless, I still have concerns where the authors provided inadequate responses to my comments and I would still recommend major revisions to address them.*

*First, regarding the datasets used for the analysis, the authors explained that they are not using published NSIDC datasets to complete the period 2000-2012 because the resolution is too coarse (1km) compared to their 500m sampling resolution. I would like to point out that if their sampling resolution is 500m, the true spatial resolution of displacement maps obtained from MODIS are certainly much coarser. Typically, the native resolution of MODIS is 250m, which would mean that cross-correlation from PyCorr is done on 2x2 pixels subimages to match the final reported resolution, which seems unrealistic, if not impossible. In addition, this NSIDC time-series https://nsidc.org/data/NSIDC-0545/versions/1 provided maps in 2000, 2006-2013 at a sampling resolution of 450 m. While I agree that the period 2001-2005 is not covered in the dataset and that MODIS could be useful on the ice shelf to bring additional information, I would still think that this external dataset would have proven useful.*

**Author response:** We appreciate the reviewer's efforts to recommend good datasets that could improve our analysis. We agree that the InSAR time series available from NSIDC has value. However, it is not just the resolution that is the problem, but also the data coverage.

First of all, the interpretation of our processing resolution presented here is incorrect. Feature tracking does rely on matching subimages, often referred to as reference chips, from the first image to search chips defined in the second image. However, the size of the reference chip does not determine the resolution of the final product. The reference chip group of pixels is unique to each flow vector determination, and the final resolution depends on the increment (or grid spacing) at which each displacement vector calculation is made. This is similar to the determination of other common neighborhood operations such as gradients or surface slope, at a posting spacing up to the pixel resolution of the input raster. This is outlined in Fahnestock et al. (2016) in reference to PyCorr specifically, and the general method is described in many other papers and remote sensing textbooks. We calculated our PyCorr image correlations every two pixels in the input MODIS images, yielding a 500 m spatial resolution.

Second, the problem is not just resolution, but also data coverage. The dataset recommended here is available in 2000, 2002, and 2006-2012, but there is no or almost no coverage on the TEIS in 2000, 2002, 2011, and 2012, and some missing coverage in the middle of the shelf in 2006. As the remaining velocity maps are only available during years where our data provide complete coverage of the shelf, and as our conclusions are already strongly supported by our data, we do not believe that additional datasets are necessary.

*Reviewer comment: Thus, even after the review, I am not convinced by the MODIS result, which also casts doubt on the strain rate analysis. It is clear that the results obtained from MODIS are wrong in many places and remain fuzzy and patchy in others. I have attached the 2006 velocity map at 450 m resolution published by NSIDC, using the exact same color coding between 0 and 3 m/day that in Figure 4, so that the maps can be compared with the result from 2005-2006 in Figure 4 from the authors. The 2000-2001 and 2005-2006 speed maps from Alley et al. indicate that the ice is not moving upstream of the GL but also near the GL on floating ice. One clear example of this issue is the main ice tongue not moving near the GL in 2005-2006 while it should be moving at 2 km/yr or more than 5 m/day. The resulting strain maps calculated from these erroneous displacement maps are therefore also erroneous. Abnormally high strain rates are visible near the GL in 2000-2001 and 2005-2006, but still are highlighted (dashed black ovals in figure 4 for example) as features showing the evolution of strain on TEIS. This sensor-dependent issue is also evident in Figure 3 near the GL and on the grounded ice where the patterns of speed and strain change as soon as the authors include Sentinel1 results. As mentioned, this also raises questions about the quality of data obtained further from the grounding line and the interpretation that can be made from these very uncertain measurements.*

**Author response:**
Feature tracking is well-established and has been repeatedly validated. We are using the same software as used in the GO-LIVE project (Fahnestock et al. 2016). Furthermore, many studies (e.g. Haug et al. 2010, Chen et al. 2016, Greene et al. 2018) have demonstrated the reliability of feature tracking on MODIS data specifically on Antarctic ice shelves, where large features are successfully tracked, even with the relatively low spatial resolution of MODIS data. We have added mentions of these citations in lines 129-131 in the manuscript. We have quantified both the absolute error in individual image-pair correlations and the empirical uncertainties associated with our combined images. Our conclusions rest upon signals that fall well outside the range of this uncertainty.

However, Reviewers 2 and 3 both point out that the incorrect MODIS results above the grounding line (due to the fixed nature of some surface features on grounded ice, e.g., undulations induced by ice-bedrock interaction) are distracting, which is a reasonable concern. The effect that the reviewer points out here was discussed in the previous submission in lines 104-109, 312-316, and 346-349; it is both expected and acknowledged in the manuscript. Our

conclusions do not depend on the data in these areas. Furthermore, we circled this area in Figure 4 to point out this expected discrepancy, not to use it in our analysis. However, we see the reviewers' points that a quick read or glancing through the figures without reading the text could cause a misleading interpretation of our data, and we certainly wish to avoid that. We therefore agree that it is best to mask these areas out in the MODIS data.

Unfortunately, implementing this is not easy, which is why we had originally allowed these data to remain. As shown in Figure 1, the grounding line on the TEIS changes rapidly and dramatically through our time period of analysis. However, there are no annual mappings of the grounding line published, nor are there enough data available to produce annual mappings. Since our time series is annual, we cannot accurately mask at the grounding line. We have therefore taken a conservative approach, masking the MODIS data above the 2000 InSAR grounding line (Rignot et al. 2016) for data between 2000 and 2004, above the ~2004 ASAID grounding line (Bindschadler et al. 2011) for data between 2004 and 2011, above the 2011 MEaSUREs grounding line (Rignot et al. 2016) for data between 2011 and 2017, and above the 2017 InSAR grounding line (Millillo et al. 2019) for the rest of the record. In addition, we imposed a minimum speed of 0.4 m/day for all MODIS data, which is more than twice the minimum velocity shown in the upstream Landsat data in the latter part of our record. We described this in lines 113-119 in our manuscript. These measures are likely to remove more data than strictly necessary. However, because we cannot accurately identify annual grounding line change, we take a conservative approach in order to increase confidence in the remaining data. In addition, in response to both this review and Reviewer 3's suggestions to be more aggressive in our filtering, we increased the size of our median filter from 3x3 to 7x7, and increased the minimum threshold for correlation strength and difference in correlation strength from neighboring options.

The reviewer is concerned that the MODIS data are both "fuzzy and patchy," and encourages us to more rigorously filter the data. We have carried out this more rigorous filtering as described above. However, these actions will, by their very nature, make the data both more fuzzy, as it suppresses variability and smooths the data, and patchy, as it eliminates data coverage in some areas during the first few years of our record, when data availability is limited. There is no valid way to address all these concerns with the available data. However, **these changes did not alter any of our conclusions.** The remaining data in our view can be used with confidence, including the area above the grounding line. We believe these data are sufficient for the needs of this paper and for future work with the dataset.

***Reviewer comment:*** *At minimum, the authors should try to filter out the maps from 2000 and 2012 for spurious measurements, and state the real resolution of their mapping with MODIS. With the filtered and correct speed maps, they could then correct the lagrangian elevation changes, correct the strain rate calculation by updating Figure 2, Figure 3 and Figure 4, especially near the GL and on grounded ice, and adjust, if needed, their interpretation of the*

*evolution of the strain on TEIS. I believe that it is important that the primary data source (ice displacement) is properly and correctly established, otherwise all the other observables (strain, melt) will be wrong and misinterpreted.*

**Author response:** The resolution of the MODIS mapping that we have presented is the real resolution. We are using established methods (based on correlation strength and difference between correlation strength and nearby correlations) for filtering out spurious measurements, which were described in lines 130-134 in our previous submission, and are highlighted in lines 141-145 in our current submission. We present empirically derived uncertainties, as well as error estimates from individual image correlations, using well-established methodologies. Our conclusions rest on results that are well above these error and uncertainty measures. We believe that our filtering routines are sufficiently rigorous and scientifically appropriate for MODIS-derived velocity data, and we have clarified that many other studies have successfully used these data for velocity correlations. We additionally have modified the presentation of the data to make the figures more easily interpreted by a casual reader. Furthermore, we have demonstrated that MODIS is the best available for the full period of our study, and in many years the only available dataset to fit the needs of this study. Because of these reasons, and because our results show signals far above the noise present in the measurements, we believe our velocity data and methods are sufficient for this study and beyond.

*Reviewer comment: My second remark concerns the 1996 and 2000 grounding line (GL) from InSAR. I would strongly suggest adding it in the manuscript. Not adding the 1996 GL because it does not overlap their analysis is questionable, but it is their choice. Nevertheless the InSAR grounding line has also been mapped in 2000 which is part of their analysis. In addition, the 2004 grounding line seems to be collected between 1999 and 2003 rather than 2004 as stated here https://www.usap-dc.org/view/dataset/609489 or https://nsidc.org/data/nsidc-0489. Please clarify at which date or period (or the source if not ASAID from Bindschadler et al. (2011)) is their 2004 GL in Figure 1*

**Author response:** The dates mentioned in the NSIDC description of the Bindschadler et al. (2011) grounding line dataset do indeed make the data collection years unclear, but the description in the Bindschadler at al. (2011) manuscript is much clearer. The dataset is derived from Landsat imagery from 1999-2003, and from ICESat laser altimetry from 2003-2009. The 2004 date we use is the center point of that time period. We have clarified this in the text in lines 236-237. The 1996 grounding line remains outside our study period, and we see no relevance in adding it to the already busy figures. The 2000 grounding line is discontinuous and covers only half of our study area as shown in Figure 1, which is why we had originally declined to use it. However, we have now added it to Figure 1, and used it for masking the MODIS data above the grounding line as described above.

*In the figure above, I plotted the GL from ASAID (Bindschadler et al. 2011) as a thick dark grey line and from InSAR as thin colored lines. It appears not only that the ASAID GL seems off by several kilometers in many places but also that having the complete evolution from 1996 to 2017 does not seem to be less relevant to their analysis.*

**Author response:** As we have shown that the Bindschadler et al. (2011) grounding line is not precisely coincident in time with any of the InSAR grounding lines, it would be surprising if it exactly matched any of them in this rapidly evolving region.

**Reviewer comment:** *Finally, the authors mentioned in their responses that the lagrangian analysis for calculating elevation changes requires a gridded dataset, which I believe is not true. There are many studies published using lagrangian approach on non-gridded datasets such as ERS, ENVISAT, IceSAT or CryoSAT (Adusumilli et al. 2020; Moholdt et al. 2014), some of them would have had sufficient resolution to capture the TEIS evolution (Gourmelen et al. 2017).*

**Author response:** The reviewer makes a good point, and we certainly should have been clearer in our explanation. We meant that Lagrangian approaches require *interpolation* of some sort, which is often achieved through the use of a gridded dataset. A high-resolution DEM such as REMA will have the highest accuracy in a Lagrangian analysis, as interpolation is minimized and is therefore ideal to accurately infer changes on the ice-shelf surface. As TEIS has extremely complex basal topography, particularly in extensive heavily crevassed areas, we are convinced this is the only method to give us the results we need for this study. Adusumilli et al. (2020) and Gourmelen et al. (2017) began with an altimetry dataset, but created a gridded dataset from it, which is functionally the same approach that we used, but the resulting reference grid is at a much lower resolution than the REMA grid we used (8m). Moholdt et al. (2014) used a "nearest neighbor" approach, where the nearest altimetry point to an advected location is used to calculate the elevation difference. In another example, Sutterley et al. (2019) was similar but used a "triangulated irregular network" approach with airborne altimetry measurements. In other words, the locations were interpolated rather than the surface elevation value. This is often the only reasonable option when data are sparse, but it again will result in a much lower resolution/spatial coverage and lower accuracy result than the approach we have used with REMA. Furthermore, REMA has become the community's standard product for ice-surface elevation and we certainly see value in pointing out the changes that have occurred since its release in 2018.

**Works cited in response to Reviewer 2:**

Adusumilli, S., Fricker, H.A., Medley, B. *et al.* Interannual variations in meltwater input to the Southern Ocean from Antarctic ice shelves. *Nat. Geosci.* **13,** 616–620 (2020). https://doi.org/10.1038/s41561-020-0616-z

Bindschadler, R., Choi, H., Wichlacz, A., Bingham, B., Bohlander, J., Brunt, K., Corr, H., Drews, R., Fricker, H., Hall, M., Hindmarsh, R., Kohler, J., Padman, L., Rack, W., Rotschky, G., Urbini, S., Vornberger, P. and Young, N.: Getting around Antarctica: New high-resolution mappings of the grounded and freely-floating boundaries of the Antarctic ice sheet created for the International Polar Year, The Cryosphere, 5(3), 569–588, https://doi.org/10.5194/tc-5-569-2011, 2011.

Chen, J., Ke, C., Zhou, X., Shao, Z., and Li, l. "Surface velocity estimations of ice shelves in the northern Antarctic Peninsula derived from MODIS data." *Journal of Geographical Sciences*, 26: 243-256, (2016).

Fahnestock, M., Scambos, T., Moon, T., Gardner, A., Haran, T. and Klinger, M., 2016. Rapid large-area mapping of ice flow using Landsat 8. *Remote Sensing of Environment*, *185*, pp.84-94. https://doi.org/10.1016/j.rse.2015.11.023

Greene, C.A., Young, D.A. Gwyther, D.E., Galton-Fenzi, B.K., and Blankenship, D. "Seasonal dynamics of Totten Ice Shelf controlled by sea ice buttressing." *The Cryosphere*, 12: 2869-2882, (2018). https://doi.org/10.5194/tc-12-2869-2018

Gourmelen, N., Goldberg, D. N., Snow, K., Henley, S. F., Bingham, R. G., Kimura, S., … van de Berg, W. J. (2017). channelized melting drives thinning under a rapidly melting Antarctic ice shelf. *Geophysical Research Letters*, 44, 9796– 9804. https://doi.org/10.1002/2017GL074929

Haug, T., Kääb, A., and Skvarca, P. "Monitoring ice shelf velocities from repeat MODIS and Landsat data – a method study on the Larsen C ice shelf, Antarctic Peninsula, and 10 other ice shelves around Antarctica." *The Cryosphere*, 4: 161-178, (2010). https://doi.org/10.5194/tc-4-161-2010.

Milillo, P., Rignot, E., Rizzoli, P., Scheuchl, B., Mouginot, J., Bueso-Bello, J. and Prats-Iraola, P.: Heterogeneous retreat and ice melt of Thwaites Glacier, West Antarctica, Sci Adv, 5(1), eaau3433, https://doi.org/10.1126/sciadv.aau3433, 2019.

Moholdt, G., Padman, L., and Fricker, H. A. (2014), Basal mass budget of Ross and Filchner-Ronne ice shelves, Antarctica, derived from Lagrangian analysis of ICESat altimetry, *J. Geophys. Res. Earth Surf.*, 119, 2361– 2380, doi:10.1002/2014JF003171.

Rignot, E., Mouginot, J. and Scheuchl, B.: MEaSUREs Antarctic Grounding Line from Differential Satellite Radar Interferometry, Version 2, NASA National Snow and Ice Data Center Distributed Active Archive Center, https://doi.org/https://doi.org/10.5067/IKBWW4RYHF1Q, 2016.

Sutterley, T. C., Markus, T., Neumann, T. A., van den Broeke, M., van Wessem, J. M., and Ligtenberg, S. R. M.: Antarctic ice shelf thickness change from multimission lidar mapping, The Cryosphere, 13, 1801–1817, https://doi.org/10.5194/tc-13-1801-2019, 2019.

**Reviewer 3:**

*Reviewer comments in italics;* Author responses in normal font

***Reviewer comment:*** *As a 3rd pair of eyes as a new reviewer, I have focused mostly on reading the arguments of previous reviewers while simultaneously having a fresh look at the paper.*

*In this review analysis I agree with the previous reviewers that i) the paper is important and ii) the results are sound. I do however also agree with R2 that many of the data sets and results are presented as such without being extremely careful in their interpretation which partly weakens the credibility of the results. If the same results are presented more rigorously, the paper would be more convincing. Moreover, I think many of the figures can be improved to increase readability.*

*I also don't think the author's response properly addresses the weak points identified by R2 and another major revision might be needed.*

**Author response:** We thank the reviewer for taking the time to add a third perspective to our paper. We have done our best to address all comments from Reviewer 2 and from this review to enhance the presentation of our results while maintaining the scientific quality and integrity of our work. We have taken several steps to improve the presentation of the MODIS data, as outlined in the comment responses below, including masking the data upstream of the grounding line, imposing a minimum speed for MODIS correlations, increasing the size of our median filter, and increasing the size of the sample boxes that are averaged and plotted in Figures 2 and 5. In addition, we have interpolated the data in Figure 7 to make the data presentation clearer, and we have clarified our text to emphasize that dynamic ice thinning/thickening was already included in our basal melt rate equation (Equation 3). Finally, we have done our best to answer each minor comment and update the figures and clarify the text accordingly.

***Reviewer comment:*** *MAJOR COMMENTS:*
*- I agree with the authors that the MODIS velocity data set is an important data set that can fill in several gaps that cannot be obtained by other existing data sets, it still seems that the MODIS data product is far from perfect with several patches of speed and direction that seem outliers and therefore incorrect (both in Fig 4 + 9). This makes the interpretation of derived strain rates, flow directions dubious. Etc. I would recommend to do a much more rigorous velocity data filtering / post-processing to identify the source of these errors and mitigate, filter it out.*

**Author response:** We have taken several steps to improve the quality of the MODIS data. First, we have done our best to manually mask out the MODIS data upstream of the grounding line, where we do not expect it to be accurate. We initially included these data because annual

grounding lines are not available for masking and our analysis focuses only on the floating ice shelf. However, we accept the reviewer's point (also expressed in a comment below) that these data can be distracting and confusing, particularly if one were to only look at the figures in the paper without reading the main text. We therefore masked the MODIS data above 2000 InSAR grounding line (Rignot et al. 2016) for data between 2000 and 2004, above the ~2004 ASAID grounding line (Bindschadler et al. 2011) for data between 2004 and 2011, above the 2011 MEaSUREs grounding line (Rignot et al. 2016) for data between 2011 and 2017, and above the 2017 InSAR grounding line (Millillo et al. 2019) for the rest of the record. In addition, we imposed a minimum speed of 0.4 m/day for all MODIS data, which is more than twice the minimum velocity shown in the upstream Landsat data in the latter part of our record. We describe this in lines 113-119 in our manuscript. These are both conservative actions, which are likely to remove more data than strictly necessary. However, because we cannot accurately identify annual grounding line change, we would rather take a conservative approach in order to have as much confidence as possible in the remaining data.

Our original data were filtered using a 3x3 median filter and by imposing minimum threshold values on the feature correlation strength and on the difference in correlation strength between successful correlations and neighboring options. These are well-established and acceptable methods for filtering velocity fields produced through feature tracking (e.g. Fahnestock et al. 2016). In response to the reviewer's concerns, we have increased the size of our median filter to 7x7 and increased the minimum thresholds for correlation strength and correlation difference. In addition, we increased the size of the squares we use for average values in Figures 2 and 5 from 3x3 km to 5x5 km in order to reduce any possible effects of noise in our analysis. We note that the more rigorous filtering has caused some data gaps early in the record, particularly right at the grounding line, so we had to shift our grounding line box to an alternative location.

Overall, these measures have increased the quality of the data above the grounding line, increased the size of patches in the first few years of the record that lack any data, and decreased the amount of detail in our long-term velocity record. While this does have some disadvantages, we believe that it overall improves our velocity record. Furthermore, these adjustments did not alter any of our conclusions, as they were well outside any noise in the record before this additional filtering, and are robust to these changes.

*Reviewer comment: - Once the TEIS-TWIT shear margin disintegrated, I don't think it is fair anymore to calculate the strain rate and/or interpret it. This has become a loose connection of icebergs that do no longer strain each other.*

**Author response:** Masking out data in the shear margin presents a similar difficulty to masking out data above the grounding line: we do not have any objective way of delineating where coherent ice-shelf ends and mélange begins. However, in this case the data calculated in the

shear margin are not necessarily incorrect, they just represent something different. Several studies have shown that ice mélange, such as the iceberg debris in this shear margin, plays a role in affecting backstress at tidewater glacier and ice-shelf fronts (e.g. Pollard et al. 2018; Cassotto et al. 2015). The strain rates calculated within this mélange are related to the amount of backstress it imposes on the intact ice shelf and upstream glacier. While this area is not directly part of our analysis in this paper, and therefore does not affect the results, we feel that removing it is not necessary and likely to be detrimental to future use of the dataset.

*Reviewer comment: - L310-315 and Fig.4: This is a nice illustration of the dubious results due to the low confidence MODIS velocity field (e.g. with land ice flowing upstream in 2001-2002, 2005-2006) and hence potential dubious interpretation of the strain rates. I think such low quality data should be removed.*

**Author response:** We agree, and have now removed these data, as described above. We do note that these data have never been part of our analysis.

*Reviewer comment: - Fig.5: what is the reason for the JJA2018 dip in mid-shelf long strain rate? I guess it noise (also visible in Fig.6), but that really makes me doubt the validity of the uncertainty intervals in Fig.5 (and of the uncertainty overall), especially as -8e-5 is way outside of the strain rates in Fig. 2 as well.*

**Author response:** That's a good question. This anomaly remains, even after our more aggressive filtering, although it is somewhat subdued after increasing the size of the boxes we are using for averaging. While it's possible it is noise, it is also possible that this is a real feature in response to an external forcing, such as the removal of fast ice during the spring season, which has been shown to influence the velocity of other ice shelves (e.g. Greene et al. 2018). A detailed analysis of the external factors that influence seasonal strain rates is beyond the scope of this manuscript, but we hope this will provide the foundation for future work. We have added a mention of this in lines 415-416 in our manuscript.

*Reviewer comment: - Given the large salt/pepper effect in Fig.6, I doubt what the added value of Fig.5 is as it only showing a constant + potential salt and pepper effects*

**Author response:** Based on this comment, we have clarified the text concerning Figures 5 and 6. Figure 5 shows averages from our lower resolution, long-term record. Figure 6 shows both our long-term record and a higher resolution record derived only from Sentinel-1. The salt and pepper effect is in the Sentinel-1 record, not in our long-term record, which is much smoother because it has the benefit of averaging data from several sources (Sentinel-1, Landsat, and MODIS). Figure 5 does not suffer from the salt and pepper effect. We have clarified this throughout section 3.2 in the text. In addition, as noted above, we increased the size of the boxes

used for averaging in Figure 5 from 3 km x 3 km to 5 km x 5 km, which reduces the possibility that noise is significantly influencing those averages.

*- L404-411: this is another dubious result. If the REMA analysis is incorrect here, why would it be correct elsewhere? Especially as it shows several areas that are remarkable but never mentioned. The positive basal melt rates for example could also indicate dynamic ice convergence etc. Just saying that REMA is wrong there (and only there) is a weak argument.*

**Author response:** The REMA mosaic is a gridded dataset, and it is reasonable to expect that discrepancies are more likely at seams in a gridded dataset. While this may not be the whole explanation, we feel that it is an important aspect to mention. We have edited our text in lines 464-466 and 595 to qualify this statement and offer other explanations. However, this anomaly is not the result of dynamic ice convergence, because that is something we have explicitly accounted for in our calculations. The second term on the left-hand side in Equation 3 is the ice thickness multiplied by the vertical strain rate, which is ice thinning/thickening. We solve this conservation equation for basal melt rate alone, so we have already removed ice thinning/thickening from our results according to our best available data and methodology. We have clarified that we have accounted for this effect in lines 294-297 and 583.

**Reviewer comment:** *- Basal melt rate analysis: the authors attribute all changes to basal melting, but don't quantify the role of convergence/divergence or changes in firn air content etc. I think these should also be quantified if the role of basal melting is quantified. Given the slowdown near pinning point I would expect for example a lot of thickening due to compression, but this is never mentioned nor quantified.*

*- Based on the previous comments I agree with R2 that the analysis of basal melting should be clarified with a better quantification of the uncertainty and inclusion of the patterns of thinning/thickening due to changes in SMB or con/divergence.*

**Author response:** We have quantified convergence/divergence, firn-air content, and SMB, as explicitly stated in Equations 2 and 3. Our treatment of firn-air content is described in Equation 2, and we outlined our estimates of temporal variability and uncertainty in lines 267-274 in the first revision, and lines 282-288 in the current submission. The mass conservation in Equation 3 accounts for ice convergence/divergence in the second term on the left-hand side and surface mass balance in the first term on the right-hand side. The basal melt rates calculated from Equation 3 and shown in Figure 7 therefore represent only basal melt, according to our best available data and methodology - surface mass balance and convergence/divergence (which, when multiplied by ice thickness, estimate ice thinning/thickening) have already been removed. If the reviewer is referring to calculating ice-thickness advection, that would only be relevant in an Eulerian analysis. The Lagrangian method uses the total derivative following ice-parcel flow,

which means that ice-thickness advection is already accounted for by taking measurements of change over the same column of ice. This is a standard methodology; we based our analysis on the descriptions of this method in Dutrieux et al. (2014) and Jenkins and Doake (1991), but many other papers have used the same calculations.

*Reviewer comment: - L461-470: based on the apparent noise in flow direction in the MODIS only record, I think it is over-ambitious to interpret the pre-Sentinel flow directions around the pinning point.*

**Author response:** Many studies (e.g. Haug et al. 2010, Chen et al. 2016, Greene et al. 2018) have demonstrated the reliability of feature tracking for ice-flow speed and direction on MODIS data on Antarctic ice shelves. In addition, our analysis of trends around the pinning point takes into account patterns that are consistent over many years and tens of kilometers, and the observed changes are consistent with patterns observed in speed, shear-strain rate, and visible changes in surface features in the shear zone. With these considerations and our own interpretations revealing very little noise around the pinning point, we find it reasonable that conclusions may be drawn from these patterns.

*Reviewer comment: - L 475-480: is the development of these rifts not the result of compression that result in transverse fractures?*

**Author response:** This section analyzes the observation of increased concentration of strain over time along individual rifts; we did not seek to provide a specific reason for rift formation in a fracture mechanics framework, which would be an appropriate topic for future work. However, as the rifts in question are not transverse to the axis of compression, we find it more likely that shear stresses are responsible for their initiation/propagation.

*Reviewer comment: MINOR COMMENTS:*
*- Fig.2, 3, 4, 6: axes, labels, text is too small to read*

**Author response:** We have increased the text size in all of these figures.

*Reviewer comment: - Fig.2 + 5: right column is not properly aligned with the rest*

**Author response:** Thank you; we have adjusted the alignment in both figures. Note that only the bottoms of the graphs are aligned. Because Matlab formats the graphs with the exponent ($10^{-5}$) above the graphs in the right column, the y-axis is slightly shorter for each graph than in the other columns.

*Reviewer comment: - Fig.4+8: please label the different ovals so it is clear what they mean and where they are discussed. I think the caption should also summarise what is seen in the ovals as to make it the reader easier and not requiring him/her to search.*

**Author response:** We have added numbers to the ovals in Figure 4, and now describe what these are showing in the caption and in the text. We have also added descriptions of the arrows and ovals in the caption for Figure 8.

*Reviewer comment: - Fig.4: adding grounding lines and velocity outline would help the reader to orient.*

**Author response:** We have added the 2011 grounding line to the MODIS images to facilitate comparison with the speed, direction, and strain rate maps.

*Reviewer comment: - Fig.7: axes/label problem: if this figure shows surface lowering than positive values should indicate lowering and negative values should indicate uplift, which is not the case*

**Author response:** We have adjusted the axis label to read "Surface height change," which is more generalizable.

*Reviewer comment: - Fig.7 shows a clear interpolation problem which should be masked*

**Author response:** We have fixed our interpolation, and described our updated method in the caption of Figure 7 and in lines 449-450.

*Reviewer comment: - I agree with R2 that overplotting of points in Fig.7 is misleading (as we the reader is biased by the last points being plotted that hide the points below). Re-gridding to a raster (with missing values outside the transect) would be more scientifically correct.*

**Author response:** As suggested, we have regridded these data to a raster over the transect paths using an inverse distance weighting approach, which is noted in the caption in Figure 7 and in lines 449-450.

**Works cited in response to Reviewer 3:**

Bindschadler, R., Choi, H., Wichlacz, A., Bingham, B., Bohlander, J., Brunt, K., Corr, H., Drews, R., Fricker, H., Hall, M., Hindmarsh, R., Kohler, J., Padman, L., Rack, W., Rotschky, G., Urbini, S., Vornberger, P. and Young, N.: Getting around Antarctica: New high-resolution mappings of the grounded and freely-floating boundaries of the Antarctic ice sheet created for the International Polar Year, The Cryosphere, 5(3), 569–588, https://doi.org/10.5194/tc-5-569-2011, 2011.

Cassotto, R., Fahnestock, M., Amundson, J., Truffer, M., & Joughin, I. (2015). Seasonal and interannual variations in ice melange and its impact on terminus stability, Jakobshavn Isbræ, Greenland. *Journal of Glaciology, 61*(225), 76-88. doi:10.3189/2015JoG13J235

Dutrieux, P., Stewart, C., Jenkins, A., Nicholls, K. W., Corr, H. F. J., Rignot, E. and Steffen, K.: Basal terraces on melting ice shelves, Geophysical Research Letters, 41(15), 5506–5513, https://doi.org/10.1002/2014gl060618, 2014.

Fahnestock, M., Scambos, T., Moon, T., Gardner, A., Haran, T. and Klinger, M., 2016. Rapid large-area mapping of ice flow using Landsat 8. *Remote Sensing of Environment*, *185*, pp.84-94. https://doi.org/10.1016/j.rse.2015.11.023

Greene, C.A., Young, D.A. Gwyther, D.E., Galton-Fenzi, B.K., and Blankenship, D. "Seasonal dynamics of Totten Ice Shelf controlled by sea ice buttressing." *The Cryosphere*, 12: 2869-2882, (2018). https://doi.org/10.5194/tc-12-2869-2018

Jenkins, A. and Doake, C. S. M.: Ice-ocean interaction on Ronne Ice Shelf, Antarctica, Journal of Geophysical Research, 96(C1), 791–813, https://doi.org/10.1029/90jc01952, 1991.

Milillo, P., Rignot, E., Rizzoli, P., Scheuchl, B., Mouginot, J., Bueso-Bello, J. and Prats-Iraola, P.: Heterogeneous retreat and ice melt of Thwaites Glacier, West Antarctica, Sci Adv, 5(1), eaau3433, https://doi.org/10.1126/sciadv.aau3433, 2019.

Pollard, D., DeConto, R. M., and Alley, R. B.: A continuum model (PSUMEL1) of ice mélange and its role during retreat of the Antarctic Ice Sheet, Geosci. Model Dev., 11, 5149–5172, https://doi.org/10.5194/gmd-11-5149-2018, 2018.

Rignot, E., Mouginot, J. and Scheuchl, B.: MEaSUREs Antarctic Grounding Line from Differential Satellite Radar Interferometry, Version 2, NASA National Snow and Ice Data Center Distributed Active Archive Center, https://doi.org/https://doi.org/10.5067/IKBWW4RYHF1Q, 2016.